# The catalytic subunit of *Plasmodium falciparum* casein kinase 2 is essential for gametocytogenesis

Eva Hitz[1,2], Olivia Grüninger[1,2], Armin Passecker[1,2], Matthias Wyss[1,2], Christian Scheurer[1,2], Sergio Wittlin[1,2], Hans-Peter Beck[1,2], Nicolas M. B. Brancucci[1,2] & Till S. Voss[1,2 ✉]

Casein kinase 2 (CK2) is a pleiotropic kinase phosphorylating substrates in different cellular compartments in eukaryotes. In the malaria parasite *Plasmodium falciparum*, PfCK2 is vital for asexual proliferation of blood-stage parasites. Here, we applied CRISPR/Cas9-based gene editing to investigate the function of the PfCK2α catalytic subunit in gametocytes, the sexual forms of the parasite that are essential for malaria transmission. We show that PfCK2α localizes to the nucleus and cytoplasm in asexual and sexual parasites alike. Conditional knockdown of PfCK2α expression prevented the transition of stage IV into transmission-competent stage V gametocytes, whereas the conditional knockout of *pfck2a* completely blocked gametocyte maturation already at an earlier stage of sexual differentiation. In summary, our results demonstrate that PfCK2α is not only essential for asexual but also sexual development of *P. falciparum* blood-stage parasites and encourage studies exploring PfCK2α as a potential target for dual-active antimalarial drugs.

[1] Department of Medical Parasitology and Infection Biology, Swiss Tropical and Public Health Institute, 4051 Basel, Switzerland. [2] University of Basel, 4001 Basel, Switzerland. ✉email: till.voss@swisstph.ch

Malaria is responsible for more than 400,000 deaths worldwide each year, of which most are caused by *Plasmodium falciparum*[1]. Malaria parasites have a complex life cycle during which sporozoites are transmitted to humans by the bite of an infected female *Anopheles* mosquito. Sporozoites reach the liver and multiply inside hepatocytes over several days to release tens of thousands of merozoites back into the blood stream, where they invade red blood cells (RBCs) and start the intraerythrocytic developmental cycle (IDC). Once inside a RBC, the merozoite develops into a ring stage parasite, then into a trophozoite and finally into a schizont that eventually releases up to 32 merozoites to invade new RBCs. Continuous rounds of these IDCs are responsible for all malaria-related morbidity and mortality. During each IDC, a small fraction of trophozoites undergoes sexual commitment and—after completing schizogony—their ring stage progeny differentiate within 10–12 days into either male or female gametocytes instead of entering another round of intraerythrocytic multiplication. When taken up by a female *Anopheles* mosquito feeding on an infected human, mature gametocytes undergo sexual reproduction in the mosquito midgut and ultimately generate sporozoites rendering the vector infectious to the next human host.

In the past two decades, immense control efforts have led to a global reduction in malaria cases and malaria-related deaths[1]. However, this success has stagnated in many regions of the world in the last 5 years and in some areas the number of cases even increased[1]. Furthermore, it has become evident that resistance to the frontline drug Artemisinin used in combination therapies can arise, resulting in delayed parasite clearance and thus an increased transmission potential[2]. Further drawbacks include the resistance of the *Anopheles* vector to insecticides as well as behavioural changes of the mosquitoes resulting in increased numbers of infectious bites[3]. It is therefore evident that new interventions and treatments are needed to further decrease the global malaria burden in the future. In recent years, kinases have gained more attention as potential drug targets and with Glivec (Imatinib Mesylate), the first kinase inhibitor for cancer treatment has been approved by the FDA in 2001[4]. Several *P. falciparum* kinases have also been proposed as potential drug targets and MMV390048, targeting the phosphatidylinositol 4-kinase, is currently tested in clinical phase 2 trials [5–9]. Another kinase that has repeatedly been referred to as an attractive potential antimalarial drug target is the *P. falciparum* casein kinase 2 (PfCK2)[10–13].

In most eukaryotes, the CK2 protein kinase is a tetramer consisting of two catalytic alpha subunits (α and α′) and a dimer of regulatory beta (β) subunits[14]. CK2 has been described as a pleiotropic and constitutively active kinase shown to be associated with the phosphorylation of hundreds of substrates in different eukaryotes including yeast and humans [15–18]. CK2 substrates are found in various cellular compartments such as the nucleus, the Golgi apparatus, the endoplasmic reticulum, and the cytoplasm[14,19]. For instance, transcription factors and other nuclear proteins were identified as CK2 substrates or interactors in yeast[20,21]. The diversity of substrates and their numerous subcellular localisations explain the essentiality of CK2 in cell differentiation, proliferation, and apoptosis as well as in the processes of gene expression and protein synthesis[14,17,19]. Gene disruption experiments further confirmed the essentiality of CK2 in different model eukaryotes including mice and yeast[22–24]. Intriguingly, in contrast to other eukaryotic CK2 kinases *P. falciparum* PfCK2 is composed of only one catalytic alpha subunit (PfCK2α) and two regulatory beta subunits (PfCK2β1 and PfCK2β2)[25,26]. All PfCK2 subunits are expressed throughout the IDC and localise to the parasite cytoplasm and nucleus[10,27–29]. Both PfCK2 subunits were found resistant to knockout attempts suggesting they are essential for parasite viability[10,12]. Furthermore, pull-down experiments performed in

two different studies revealed that PfCK2α interacts with both β subunits[10,12] and the β subunits are able to regulate PfCK2α activity in vitro[12]. The catalytic PfCK2α subunit potentially undergoes auto-phosphorylation at several sites (e.g., Thr$^{30}$, Tyr$^{20}$) to regulate kinase activity[11–13]. The importance of Thr$^{30}$ auto-phosphorylation has been confirmed since mutation of this site prevented auto-phosphorylation and reduced kinase activity[11].

PfCK2 is likely involved in chromatin dynamics since chromatin-related proteins such as nucleosome assembly proteins, histones and members of the ALBA protein family are likely substrates of PfCK2α[10]. Nuclear proteins involved in mitosis and DNA replication as well as proteins associated with merozoite invasion, motility and post-translational modification were also identified as potential interactors and substrates of PfCK2[10]. Furthermore, *P. falciparum* merozoite invasion ligands of the reticulocyte binding-like homologue (Rh) and erythrocyte binding-like antigen (EBA) families are phosphorylated at their cytoplasmic domain by PfCK2 in vitro and these modifications seem to play an important role during the RBC invasion process[31,32]. Conditional knockdown (cKD) of PfCK2α expression in 3D7 parasites caused a lethal phenotype related to a defect in merozoite invasion, highlighting that PfCK2α is essential for the propagation of asexual blood stage parasites[32]. Hence, PfCK2 clearly has a variety of substrates and vital roles also in *P. falciparum*, which is in line with the wide range of functions attributed to CK2 in other eukaryotes. Interestingly, PfCK2α can be chemically inhibited by confirmed CK2 inhibitors including CX4945, 3,4,5,6-tetrabromobenzotriazole (TBB) and Quinalizarin in in vitro kinase assays with fifty percent inhibitory concentrations ($IC_{50}$) in the low μM to nM range[11–13]. A kinase-directed inhibitor library screen identified the kinase inhibitor Rottlerin as differentially active against PfCK2 ($IC_{50} = 7$ μM) compared to human CK2 ($IC_{50} > 20$ μM), suggesting the possibility of selectively targeting PfCK2[12]. However, whether CK2 kinase inhibitors block parasite growth in in vitro cultures and in vivo remains to be determined.

Here, we performed an in-depth functional analysis of PfCK2α in asexual blood-stage parasites and during sexual commitment and gametocytogenesis. Our data confirm the importance of PfCK2α in merozoite invasion and intra-erythrocytic development. Furthermore, we demonstrate its essentiality for gametocyte maturation. In addition, we show that out of four validated CK2 inhibitors tested in in vitro kinase inhibition assays, none shows potent activity against parasite multiplication.

## Results

**CRISPR/Cas9-mediated engineering of transgenic parasite lines to study PfCK2α expression and function**. We initially engineered two parasite lines to study PfCK2α. The first line expresses PfCK2α-GFP to investigate the subcellular localisation of PfCK2α expressed at wild-type (WT) levels. The second line expresses PfCK2α tagged with a GFP-FKBP/DD destabilization domain fusion, which allows knocking down PfCK2α expression levels by removal of the stabilizing ligand Shield-1 from the culture medium[33,34]. Because one of our aims was to test if PfCK2α has a role in regulating sexual commitment, the PfCK2α-GFP and PfCK2α-GFPDD transgenic lines were generated in NF54 parasites expressing a mScarlet-tagged version of PfAP2-G (NF54/AP2-G-mScarlet) (Brancucci et al., manuscript in preparation). The transcription factor PfAP2-G is known as the master regulator of sexual commitment[35,36] and we previously used parasites expressing GFP-tagged PfAP2-G to mark and distinguish sexually committed from asexual parasites by live-cell fluorescence microscopy[37]. Here, we used a two-plasmid CRISPR/Cas9 approach[38] to generate NF54/AP2-G-mScarlet

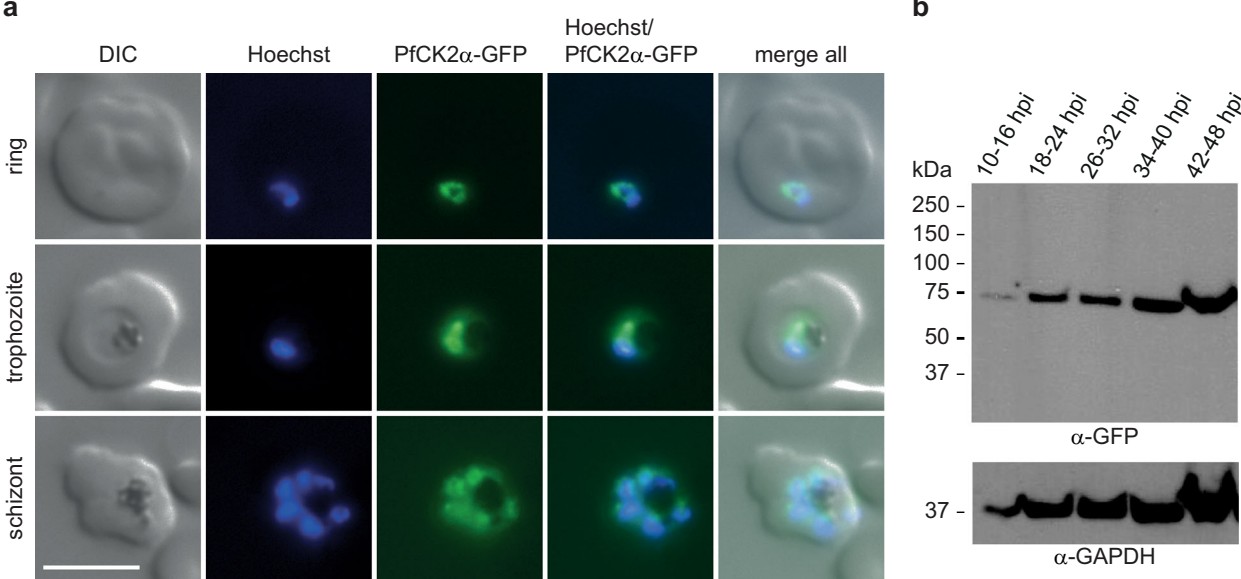

**Fig. 1 Expression and localisation of PfCK2α in NF54/AP2-G-mScarlet/CK2α-GFP parasites. a** Expression and localisation of PfCK2α-GFP in ring, trophozoite and schizont stage parasites by live cell fluorescence imaging. Representative images are shown. Nuclei were stained with Hoechst. DIC, differential interference contrast. Scale bar = 5 μm. **b** Western blot analysis showing expression of PfCK2α-GFP at several time points during the IDC. Protein lysates derived from an equal number of parasites were loaded per lane. MW PfCK2α-GFP = 66.8 kDa, MW loading control PfGAPDH = 36.6 kDa.

lines expressing C-terminally tagged PfCK2α-GFP or PfCK2α-GFPDD. Successful editing of the *pfck2α* gene and absence of the WT locus was confirmed in both lines by PCR on genomic DNA (gDNA) (Supplementary Figs. 1 and 2). We also detected integration of donor plasmid concatamers downstream of the *pfck2α* locus in at least a subset of parasites in both populations (Supplementary Figs. 1 and 2). Such undesired recombination events have previously been reported in other studies as well[39,40]. However, since the 829 bp 3′ homology region (HR) used for homology-directed repair seems to include the native terminator based on published RNA-seq data[41], expression of the modified *pfck2α* genes is likely not compromised by donor plasmid integration (Figs. 1 and 2 and Supplementary Figs. 1–3).

**Conditional knockdown of PfCK2α-GFPDD expression has no effect on asexual growth but causes a defect late in gametocytogenesis.** In the NF54/AP2-G-mScarlet/CK2α-GFP transgenic line, we investigated the expression and localisation of PfCK2α in asexual blood stage parasites by Western blot and live cell fluorescence imaging. Consistent with previous findings[10,27–29], we show that PfCK2α is expressed throughout the IDC with increased expression in trophozoites and schizonts and localises to both the nucleus and parasite cytoplasm (Fig. 1 and Supplementary Fig. 3). In the NF54/AP2-G-mScarlet/CK2α-GFPDD conditional knockdown (cKD) line, the functionality of the cKD system to regulate PfCK2α expression was confirmed by Western blot and live cell fluorescence imaging, showing a substantial reduction of PfCK2α protein levels upon removal of Shield-1 (−Shield-1) (Fig. 2a and Supplementary Fig. 2).

To identify a potential effect of the depletion of PfCK2α on asexual parasite growth, we performed multiplication assays comparing parasites cultured in presence or absence of Shield-1 (±Shield-1) over two generations by measuring fluorescence intensity of SYBR Green-stained parasites using flow cytometry. We did not observe a significant difference in the multiplication rates of PfCK2α-GFPDD-depleted parasites (−Shield-1) in comparison to the isogenic control population (+Shield-1) (Fig. 2b and Supplementary Fig. 4). This finding is in contrast to those

obtained by Tham and colleagues, who observed a dramatic multiplication defect upon conditional depletion of PfCK2α in parasites expressing PfCK2α C-terminally tagged with a triple hemagglutinin (HA) tag fused to DD[32].

We next investigated whether reduction of PfCK2α expression levels affects the ability of parasites to undergo sexual commitment and/or sexual development. To study the potential effect of PfCK2α depletion on sexual commitment, we split synchronous NF54/AP2-G-mScarlet/CK2α-GFPDD ring stage cultures at 0–6 h post invasion (hpi) and Shield-1 was removed from one of the paired cultures. After 18 h (18–24 hpi), we induced sexual commitment using serum-free medium (−SerM)[37]. Parasites cultured on −SerM medium supplemented with 2 mM choline chloride (−SerM/CC), a metabolite known to repress sexual commitment[37], were used as control. Sexual commitment rates were determined in the ring stage progeny by quantifying the proportion of PfAP2-G-mScarlet-positive parasites among all infected RBCs (iRBCs) identified by Hoechst staining. When comparing parasites cultured in presence or absence of Shield-1, no significant difference in sexual commitment rates was observed in either of the two medium conditions (−SerM and −SerM/CC) (Supplementary Fig. 5). We next monitored gametocyte maturation of sexually committed ring stage parasites cultured in presence or absence of Shield-1 over 11 days by visual inspection of Giemsa-stained thin blood smears. To perform these experiments, sexual commitment was induced in parasites previously split (±Shield-1) using −SerM culture medium as described above. The asexual/sexual ring stage progeny was subsequently maintained in culture medium supplemented with serum (+SerM) containing 50 mM N-acetyl-D-glucosamine (GlcNAc) to eliminate asexual parasites and obtain pure gametocyte populations[42]. Whereas the morphology of stage I–IV gametocytes (day 2 to 9) was comparable between NF54/AP2-G-mScarlet/CK2α-GFPDD parasites cultured in presence or absence of Shield-1, most PfCK2α-GFPDD-depleted gametocytes failed to develop into mature stage V gametocytes (day 11), even after prolonged periods of observation (Fig. 3a, b). Closer assessment of gametocyte morphology based on three independent gametocyte maturation assays revealed that from day 8 onwards the

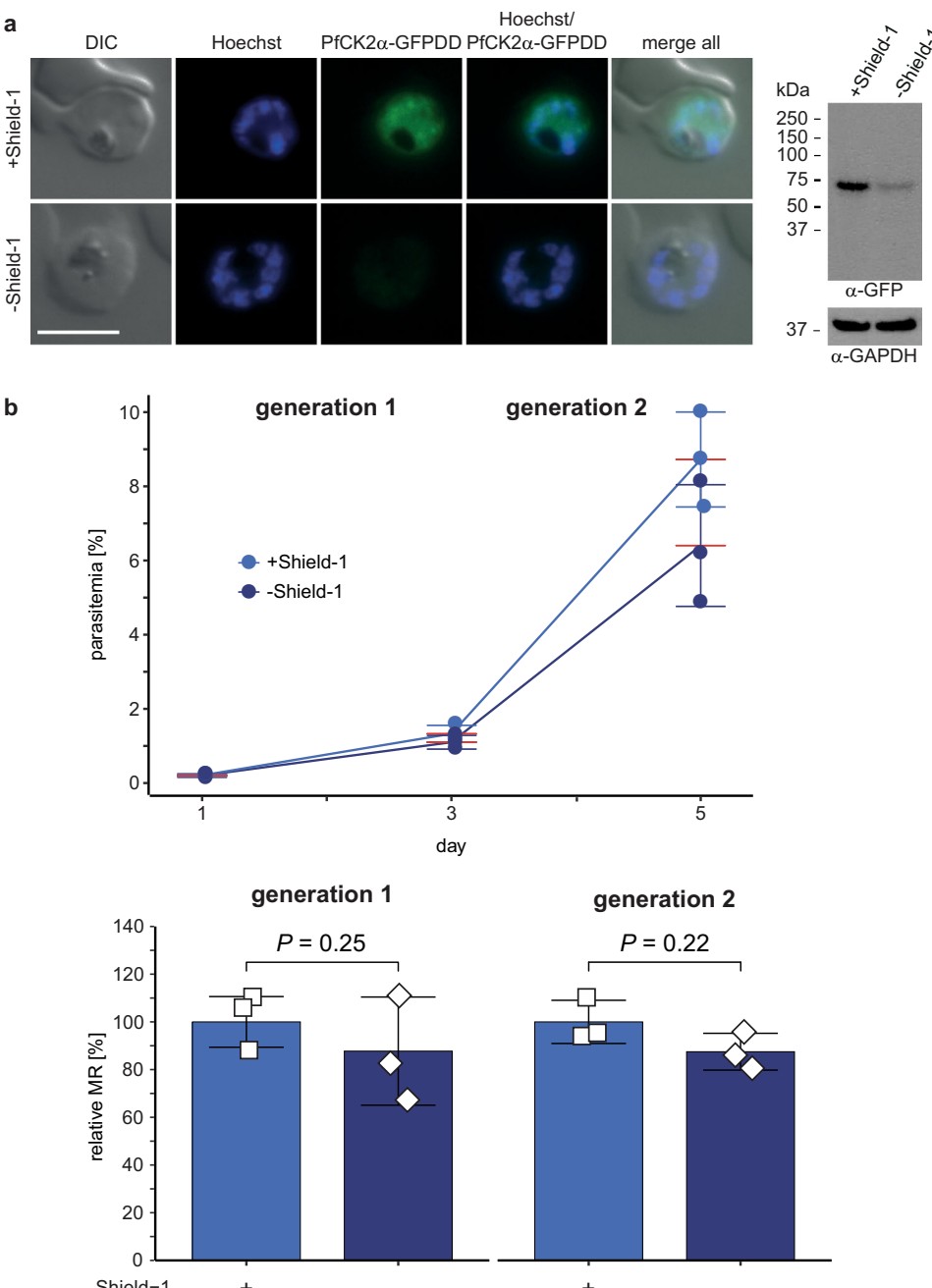

**Fig. 2 PfCK2α depletion in NF54/AP2-G-mScarlet/CK2α-GFPDD parasites has no effect on asexual parasite multiplication. a** Expression of PfCK2α-GFPDD in schizonts cultured in presence (+Shield-1) and absence (−Shield-1) of Shield-1 by live cell fluorescence imaging and Western blot analysis. Parasites were split (±Shield-1) 40 h before sample collection. Representative images are shown. Nuclei were stained with Hoechst. DIC, differential interference contrast. Scale bar = 5 μm. For Western blot analysis, protein lysates derived from an equal number of parasites were loaded per lane. MW PfCK2α-GFPDD = 78.7 kDa, MW loading control PfGAPDH = 36.6 kDa. **b** Flow cytometry data showing the increase in parasitaemia (top) and parasite multiplication rates (bottom) in two subsequent generations of NF54/AP2-G-mScarlet/CK2α-GFPDD parasites cultured under protein-degrading (−Shield-1; dark blue) and protein-stabilizing (+Shield-1; light blue) conditions. Parasites were split (±Shield-1) at 0–6 hpi, 18 h before the first measurement (day 1). The means ±s.d. (error bars) of three biological replicates are shown. Data points for individual replicates are represented by coloured circles (top) or open squares (bottom). Differences in parasite multiplication rates were compared using a paired two-tailed Student's t test (P values are indicated above the graphs). MR, multiplication rate.

morphology of PfCK2α-GFPDD-depleted gametocytes changed considerably, resulting in stage IV-type cells with elongated and pointy tips that did not progress further to adopt the typcial stage V morphology observed for +Shield-1 control gametocytes on day 11 (Fig. 3b). Western blot analysis confirmed the efficient depletion of PfCK2α in day 11 gametocytes cultured in absence of

Shield-1 (Fig. 3c and Supplementary Fig. 5). To obtain further insight into the functionality and viability of PfCK2α knockdown gametocytes we performed microsphiltration experiments. Microsphiltration experiments allow comparing differences in cell rigidity based on cell retention rates in microsphere filters that represent an artificial spleen system[43,44]. Previous research

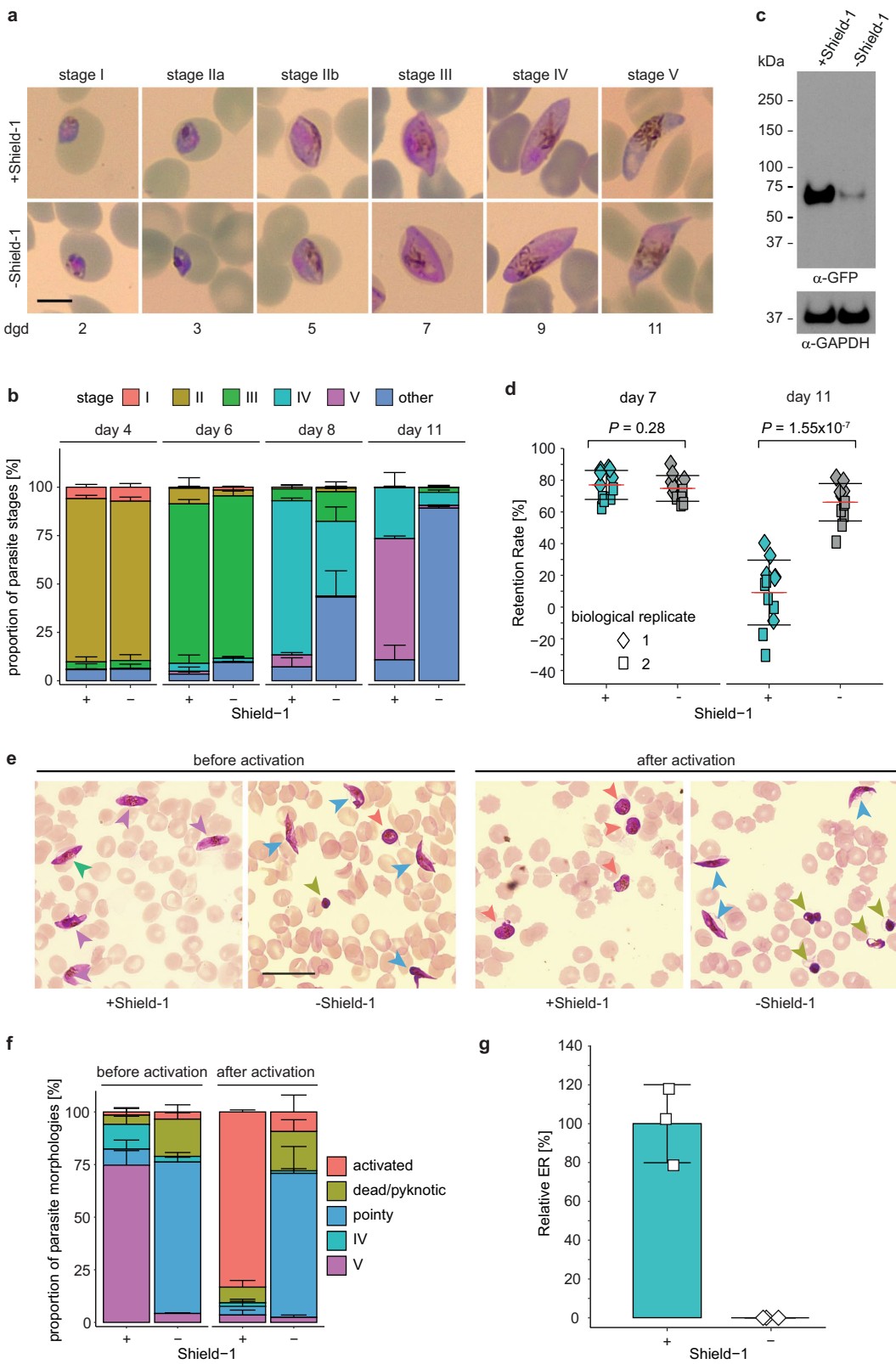

has shown that stage I–IV gametocytes are more rigid than mature gametocytes and sequester in tissues such as the bone marrow and spleen and are thus absent from the blood circulation[43,45–47]. In contrast, mature stage V gametocytes become highly deformable, which likely allows them to leave their site of sequestration and re-enter the bloodstream from where they can be taken up by a mosquito vector[43,45,46]. Here, we performed microsphiltration experiments on stage III (day 7) and stage V (day 11) gametocytes by microsphiltration[44]. Our data reveal that gametocytes with depleted PfCK2α expression (–Shield-1) do not undergo a deformability switch since gametocytes on day 11 of development were still retained on the microsphere columns like immature gametocyte stages (Fig. 3d). In contrast, in the matching control population (+Shield-1) only stage III gametocytes were retained

**Fig. 3 PfCK2α depletion in NF54/AP2-G-mScarlet/CK2α-GFPDD parasites prevents formation of mature stage V gametocytes. a** Representative images of morphologically distinct sexual stages (stage I–V) cultured in presence (+Shield-1) and absence (−Shield-1) of Shield-1 over 11 days of gametocyte development. Parasites were split (±Shield-1) at 0-6 hpi and transferred to −SerM at 18–24 hpi to induce sexual commitment[37] and subsequently cultured in presence or absence of Shield-1 until stage V gametocytes were obtained. From the ring stage progeny (asexual/sexual ring stages) until day six of gametocytogenesis, parasites were cultured in presence of 50 mM GlcNAc to eliminate asexual parasites[42]. Images were taken from Giemsa-stained thin blood smears. Scale bar = 5 μm. dgd, day of gametocyte development. **b** Proportion of gametocyte stages on day four, six, eight and eleven of gametocytogenesis in populations cultured in presence (+) and absence (−) of Shield-1, as assessed by morphological classification and counting from Giemsa-stained thin blood smears. Deformed gametocytes that could not be classified as one of the five distinct morphological stages I–V were classified as "other" (see colour code above the graph). Gametocytes were cultured as described above in panel **a**. The means ±s.d. (error bars) of three biological replicates are shown. At least 100 gametocytes were classified per sample (see Supplementary Data 1 for exact numbers). **c** Expression of PfCK2α-GFPDD in stage V gametocytes cultured in presence (+Shield-1) and absence (−Shield-1) of Shield-1 by Western blot analysis. Gametocytes were cultured as described above in panel **a**. Protein lysates derived from an equal number of parasites were loaded per lane. MW PfCK2α-GFPDD = 78.7 kDa, MW loading control PfGAPDH = 36.6 kDa. **d** Retention rates of gametocytes cultured in presence (+Shield-1; turqoise) or absence (−Shield-1; grey) of Shield-1 on day seven (stage III) and day 11 (stage V) of development as determined from microspiltration experiments[44]. Gametocytes were cultured as described above in panel **a**. The means ±s.d. (error bars) of two biological replicates performed in five to six technical replicates each are shown. Individual data points of the technical replicates from the first and second biological replicate experiments are represented by coloured diamonds and squares, respectively. Differences in gametocyte retention rates were compared using the unpaired two-tailed Student's t test (P values are indicated above the graphs). **e** Representative overview images showing gametocytes on day 14 of development before and after (10 min) the activation of gametogenesis (XA and drop in temperature) and cultured in presence (+Shield-1) or absence (−Shield-1) of Shield-1. Gametocytes were cultured as described above in **a**. Images were taken from Giemsa-stained thin blood smears. Coloured arrowheads mark the different morphological types observed according to the key provided in **f** below. Scale bar = 20 μm. **f** Proportion of gametocytes with stage IV (turquoise), stage V (purple), "pointy" (blue), activated (salmon) or dead/pyknotic (green) morphology on day 11 of gametocytogenesis in parasites cultured in presence (+) and absence (−) of Shield-1 before and 10 min after activation of gametogenesis by XA. Gametocytes were allocated to one of the five different classes of morphology based on visual inspection of Giemsa-stained thin blood smears. Gametocytes were cultured as described above in panel **a**. The means ±s.d. (error bars) of four biological replicates are shown. At least 100 gametocytes were classified per sample (see Supplementary Data 1 for exact numbers). **g** Relative exflagellation rates of gametocytes cultured in presence (+) and absence (−) of Shield-1. Gametocytes were cultured as described above in panel **a**. The means ±s.d. (error bars) of three biological replicates are shown and individual data points are represented by open squares. ER, exflagellation rate.

on the columns whereas stage V passed through efficiently as expected (Fig. 3d).

Once taken up by an *Anopheles* mosquito, male and female gametocytes become activated by several changes in the environment and consequently develop into gametes. Activation of both male and female gametocytes can be triggered in vitro by a drop in temperature and addition of xanthurenic acid (XA)[48] to the culture medium. Gametocyte activation can be observed by a change in shape since gametocytes egress from the iRBC and become spherical in a process termed "rounding up"[49]. Activation of male gametocytes further entails three rapid rounds of genome replication and production of eight male microgametes that can fuse with a female macrogamete to generate a zygote. The egress of eight motile microgametes from the iRBC is termed exflagellation and can be observed and quantified by bright-field microscopy[50,51]. Successful exflagellation was proposed as a suitable proxy for the viability of mature male gametocytes[50]. We observed that the majority of PfCK2α-depleted (−Shield-1) day 11 gametocytes with aberrant morphology (elongated, pointy tips) were unable to round up upon XA-induced activation whereas gametocytes of the matching control (+Shield-1) rounded up as expected (Figs. 3e and 3f). Consistent with this finding, PfCK2α-depleted day 11 gametocytes were also unable to exflagellate in contrast to the control population grown in presence of Shield-1 (Fig. 3g).

In conclusion, conditional depletion of PfCK2α in NF54/AP2-G-mScarlet/CK2α-GFPDD parasites has no apparent effect on parasite multiplication and on their capacity to undergo and regulate sexual commitment in response to environmental triggers. However, PfCK2α-depleted late stage gametocytes show abnormal morphology, fail to undergo the deformability switch, are severely impaired in the process of rounding up and are unable to exflagellate.

**PfCK2α is essential for asexual development.** It was previously reported that conditional depletion of PfCK2α leads to a severe

multiplication defect, where PfCK2α-depleted parasites completed schizogony but failed to give rise to ring stage progeny likely due to defective merozoite invasion[32]. Unexpectedly, we did not observe this phenotype in our NF54/AP2-G-mScarlet/CK2α-GFPDD cKD line. To address this discrepancy, we generated a conditional knockout (cKO) parasite line based on the DiCre/loxP system[52–54]. We performed two consecutive CRISPR/Cas9-based gene editing steps in NF54::DiCre parasites[52] to insert a loxP-intron/loxPint element[53] into the 5′ end and fuse a GFP coding sequence followed by a loxP element to the 3′ end of the *pfck2α* gene (NF54::DiCre/CK2α-GFP cKO line) (Supplementary Fig. 6). Successful editing of the *pfck2α* locus and absence of the WT locus was confirmed by PCR on gDNA (Supplementary Fig. 6). We again also detected integration of donor plasmid concatamers downstream of the *pfck2α* locus in a subset of parasites (Supplementary Fig. 6). In this parasite line, the *pfck2α-gfp* coding region can be excised by addition of rapamycin (RAPA) to the culture medium, which activates the DiCre recombinase to recombine the two inserted loxP sites[52]. PCR on gDNA isolated from late schizonts 40 h after treating ring stage parasites with RAPA revealed the highly efficient excision of the *pfck2α-gfp* gene (Supplementary Fig. 6) and successful depletion of PfCK2α-GFP expression was observed by Western blot analysis and live cell fluorescence imaging when compared to DMSO-treated control parasites (Fig. 4a). To assess the phenotype of PfCK2α KO parasites, we analysed parasite multiplication by visual inspection of Giemsa-stained blood smears and flow cytometry-based analysis of SYBR green-stained iRBCs (Supplementary Fig. 4). After triggering excision of *pfck2α-gfp* (RAPA-treated) in young ring stages (0–8 hpi), parasites developed similarly compared to the matching control population until the end of schizogony (Fig. 4b) but upon schizont rupture PfCK2α KO merozoites seemed unable to invade RBCs, as suggested by Tham and colleagues after knocking down expression of PfCK2α-3xHADD in 3D7 parasites[32] (Fig. 4c, d). A small subset of merozoites was still able to invade new erythrocytes, which was

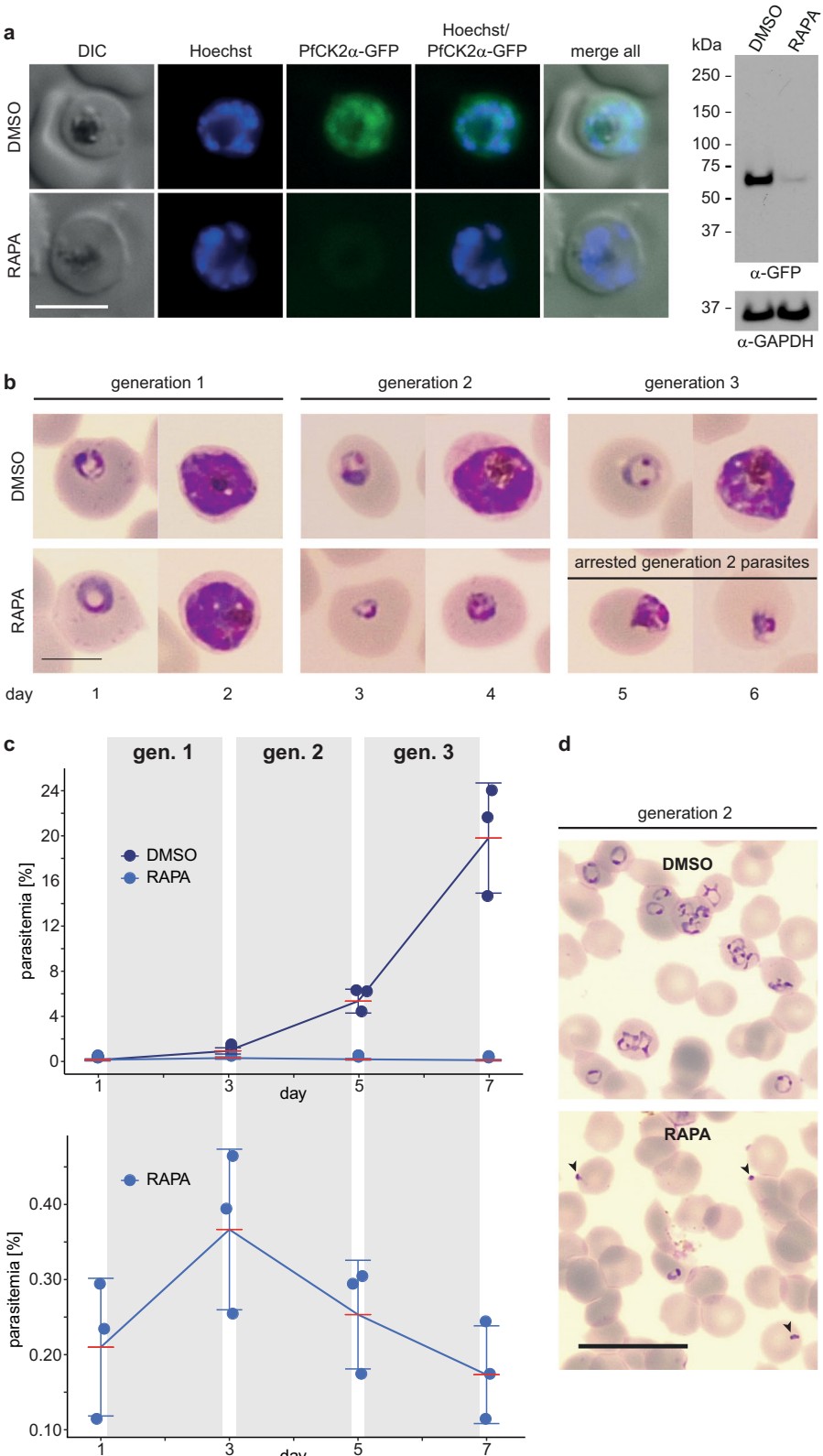

evident by a minor increase in parasitaemia in generation 2 (Fig. 4b, c, and d). Notably, however, these PfCK2α KO progeny parasites arrested at the trophozoite stage and failed to develop further (Fig. 4b, c).

To investigate the PfCK2α KO phenotype in more detail, we performed flow cytometry measurements of synchronised parasite cultures at multiple time points during the IDC. Upon

excision of *pfck2α-gfp* in young ring stages (0–4 hpi), RAPA-treated and DMSO-treated control parasites simultaneously entered the first round of genome replication at the onset of schizogony (32–36 hpi). However, PfCK2α KO parasites progressed more slowly through schizogony compared to the control population, showing a delay of approximately four hours towards the end of the IDC (44–48 hpi) (Supplementary Figs. 7

**Fig. 4 Deletion of *pfck2α* in NF54::DiCre/CK2α-GFP cKO parasites leads to a dramatic defect in asexual parasite multiplication. a** Expression of PfCK2α-GFP in NF54::DiCre/CK2α-GFP KO (RAPA) and control (DMSO) schizonts by live cell fluorescence imaging and Western blot analysis. Parasites were split and exposed for four hours to either RAPA or DMSO 40 h before sample collection. Representative images are shown. Nuclei were stained with Hoechst. DIC, differential interference contrast. Scale bar = 5 μm. For Western blot analysis, protein lysates derived from an equal number of parasites were loaded per lane. MW PfCK2α-GFP = 66.8 kDa, MW loading control PfGAPDH = 36.6 kDa. RAPA, rapamycin. **b** Representative images of NF54:: DiCre/CK2α-GFP KO (RAPA) and control (DMSO) parasites taken from Giemsa-stained thin blood smears prepared daily for six consecutive days. Scale bar = 5 μm. RAPA, rapamycin. **c** Top: Flow cytometry data showing the increase in parasitaemia over three subsequent generations in NF54::DiCre/CK2α-GFP KO (RAPA; light blue) and control (DMSO; dark blue) parasites. Bottom: Zoom-in on the marginal increase in parasitaemia observed for the RAPA-treated population. Parasites were split at 0–6 hpi and treated for four hours with either RAPA or DMSO 18 h before the first measurement (day 1). The means ±s.d. (error bars) of three biological replicates are shown. Data points for individual replicates are represented by closed circles. RAPA, rapamycin. gen., generation. **d** Representative overview images showing parasites in the progeny of RAPA- and DMSO-treated NF54::DiCre/CK2α-GFP parasites (0–8 hpi; generation 2) taken from Giemsa-stained thin blood smears. Black arrowheads indicate merozoites unable to invade new RBCs. Scale bar = 20 μm.

and 8). After an additional 10 h, schizonts had disappeared from both cultures but the ring stage progeny produced by PfCK2α KO parasites was approximately five-fold lower compared to the control population (Supplementary Fig. 7), consistent with the results presented above (Fig. 4c). To confirm that PfCK2α KO parasites are able to release merozoites, we treated late stage schizonts with the PfPKG kinase inhibitor compound 2 (C2), an experimental compound that reversibly arrests schizonts at the very end of the IDC just prior to merozoite egress[55]. After reversing the developmental arrest by washing away C2, merozoite egress was observed for both DMSO-treated control and RAPA-treated PfCK2α KO schizonts as revealed by both flow cytmetry analysis and inspection of Giemsa-stained blood smears (Supplementary Fig. 7).

In summary, our PfCK2α cKO line confirms the previously suggested essential function of PfCK2α in merozoite invasion and hence proliferation of asexual blood stage parasites[32]. We show that after knocking out *pfck2α* in ring stages, parasites progressed through schizogony more slowly but still produced and released merozoites. While the vast majority of PfCK2α KO merozoites failed to invade RBCs, a small subset of merozoites was still able to invade but were unable to develop past the trophozoite stage. Hence, PfCK2α has essential functions during the IDC in addition to its crucial role for merozoite invasion.

**PfCK2α is essential for sexual development**. Proteomics as well as transcriptomics data suggest that PfCK2α is expressed throughout sexual development[56,57]. Using the NF54::DiCre/CK2α-GFP cKO parasite line, we investigated the expression and localisation of PfCK2α-GFP during gametocytogenesis on a single cell level by live cell fluorescence imaging. Indeed, we observed PfCK2α-GFP expression throughout gametocyte maturation in all five stages in both the nuclear and cytoplasmic compartments of the parasite (Fig. 5), which is in line with our observations during the IDC (Fig. 1a and Supplementary Fig. 3). Next, we asked if knocking out PfCK2α at the onset of gametocytogenesis has more dramatic consequences for gametocyte maturation compared to knocking down PfCK2α expression in the cKD line (see above). To this end, we induced sexual commitment using −SerM medium[37], excised the *pfck2α-gfp* gene in the subsequent ring stage progeny by treatment with RAPA as described above and allowed gametocyte maturation to proceed in +SerM culture medium supplemented with 50 mM GlcNAc during the first six days. Strikingly, RAPA-treated PfCK2α KO gametocytes showed abnormal morphology during most of gametocyte development when compared to the matching DMSO-treated control (Fig. 6a) and this phenotype was more pronounced compared to the PfCK2α-GFPDD-depleted gametocytes described above (Fig. 3a). Additional investigation of this phenotype revealed a drastic defect in gametocytogenesis from stage II (day 4 to 5) onwards with most gametocytes failing to develop further (Fig. 6a and b).

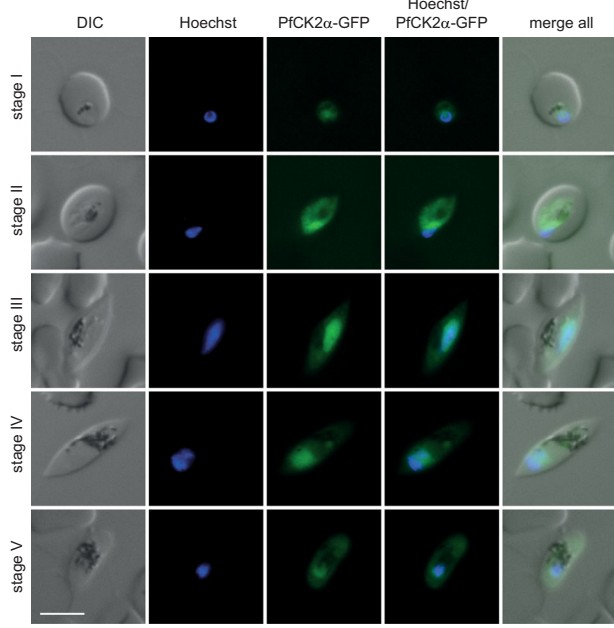

**Fig. 5 Expression and localisation of PfCK2α-GFP during gametocytogenesis (stage I–V) by live cell fluorescence imaging.** Representative images are shown. Nuclei were stained with Hoechst. DIC, differential interference contrast. Scale bar = 5 μm.

On day 11 of development most PfCK2α KO gametocytes were severely deformed, rounded up or pyknotic in appearance (Fig. 6a and b). Furthermore, the gametocytaemia in RAPA-treated populations decreased by 87% (86.4%±4.8) from day 4 to day 11, showing that a large proportion of PfCK2α KO gametocytes died during maturation. In contrast, the corresponding decrease in gametocytaemia in control DMSO-treated populations was only 44% (43.8%±1.3), which is in line with our observations from routine gametocyte cultures and can be explained by normal turnover, for instance due the lysis of iRBCs over time[50]. Microscopy of magnet-purified gametocytes on day 11 of development illustrates the drastic morphological changes and lack of PfCK2α-GFP expression in PfCK2α KO gametocytes when compared to the respective DMSO-treated control (Fig. 6c, d).

In contrast to our observations made with NF54/AP2-G-mScarlet/CK2α-GFPDD gametocytes, in which the conditional knockdown of PfCK2α-GFPDD expression does not result in major morphological changes up to day 8 (stage IV) of gametocytogenesis, knocking out the *pfck2α* gene in sexual ring stages leads to dramatic morphological changes and defective cell differentiation already early during gametocytogenesis. In

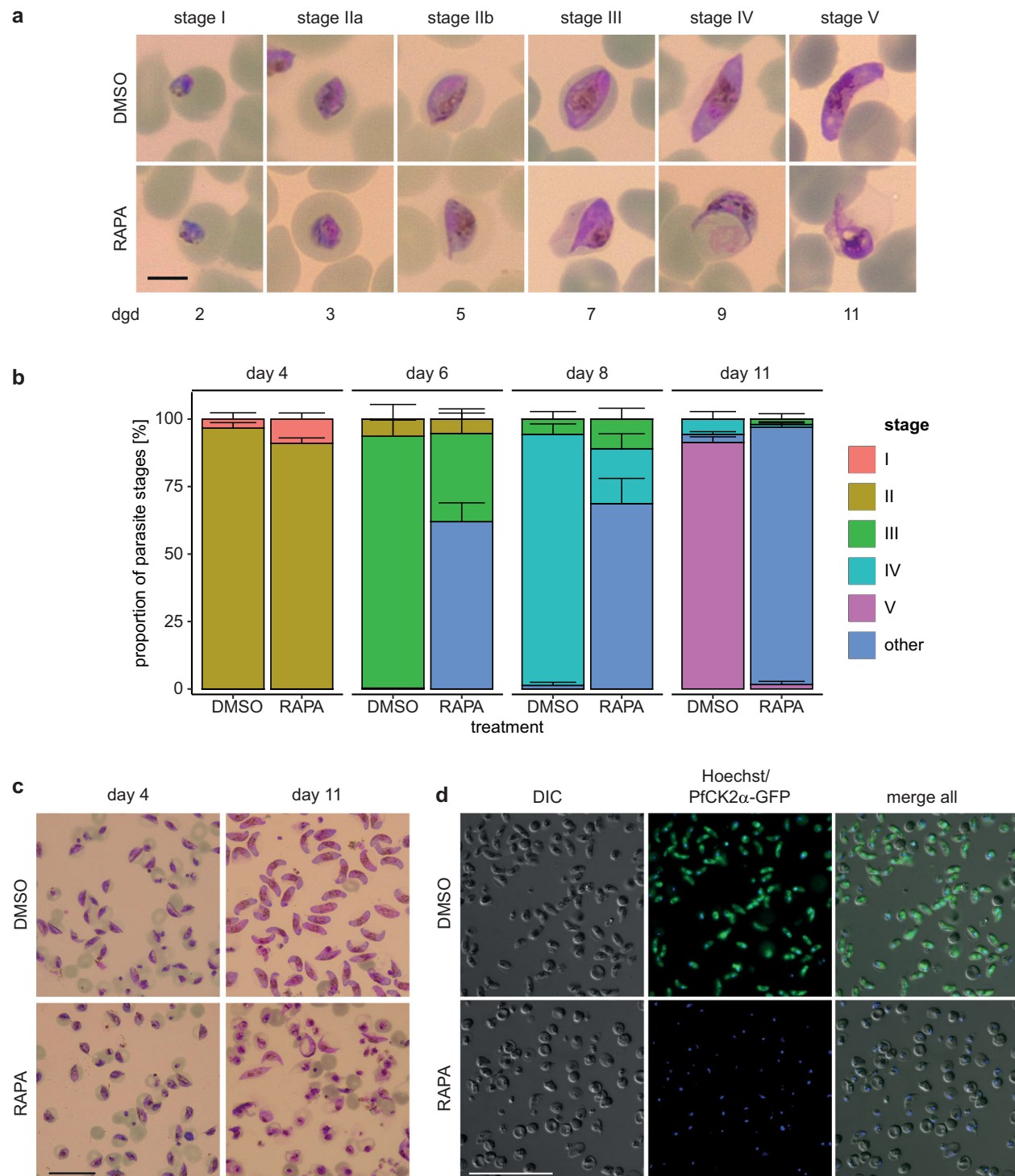

summary, these experiments demonstrate that PfCK2α is not only essential for asexual proliferation but also for sexual parasite development.

**Activity of CK2 inhibitors in reducing asexual parasite multiplication.** Whereas several studies proposed PfCK2α as an attractive drug target based on in vitro kinase inhibition assays using compounds targeting human CK2α[11–13], data on successful inhibition of parasite growth and multiplication by such compounds in cell-based assays is currently missing. To close this knowledge gap, we tested four commercially available cell-permeable and selective ATP-competitive inhibitors of human CK2α (Quinalizarin, TTP 22, TBB and DMAT) for their ability to reduce or block parasite multiplication in a [3H]-hypoxanthine incorporation assay[58]. Quinalizarin, described as one of the most potent human CK2α inhibitors, was previously reported to block the activity of recombinant PfCK2α in vitro with an $IC_{50}$ of 2 μM[11,59]. We identified an $IC_{50}$ of higher than 50 μM for Quinalizarin and at this concentration parasites still multiplied at a rate of about 65%, when compared to the drug-free control culture (Supplementary Fig. 9). Next, we tested TTP 22 that inhibits the human CK2α kinase in vitro with an $IC_{50}$ value of 0.1 μM[60]. Like Quinalizarin, TTP 22 performed poorly ($IC_{50} > 50$ μM) in

**Fig. 6 Deletion of _pfck2α_ in NF54::DiCre/CK2α-GFP cKO parasites leads to defective gametocytogenesis. a** Representative images of NF54::DiCre/ CK2α-GFP KO (RAPA) and control (DMSO) gametocytes over 11 days of development taken from Giemsa-stained thin blood smears. Parasites were induced for sexual commitment using −SerM medium[37] and the ring stage progeny was split at 0-6 hpi, treated for four hours with either DMSO or RAPA and then cultured in presence of 50 mM GlcNAc for six consecutive days to eliminate asexual parasites[42]. Subsequently, gametocytes were cultured in +SerM medium until stage V gametocytes were observed in the DMSO-treated control population. Scale bar = 5 μm. RAPA, rapamycin. dgd, day of gametocyte development. **b** Proportion of gametocyte stages on day four, six, eight and eleven of gametocytogenesis in PfCK2α-GFP KO (RAPA) and control (DMSO) populations as assessed by morphological classification and counting from Giemsa-stained thin blood smears. Deformed gametocytes that could not be classified as one of the five distinct morphological stages I–V were classified as "other" (see colour code above the graph). Gametocytes were cultured as described above in panel **a**. The means ±s.d. (error bars) of three biological replicates are shown. 100 gametocytes were classified per sample. **c** Representative overview images showing magnet-purified gametocytes on day 4 and day 11 of development in PfCK2α-GFP KO (RAPA) and control (DMSO) gametocytes. Images were taken from Giemsa-stained thin blood smears. Gametocytes were cultured as described above in panel **a**. Scale bar = 20 μm. RAPA, rapamycin. **d** Representative live cell fluorescence images showing an overview of magnet-purified NF54::DiCre/CK2α-GFP KO (RAPA) and control (DMSO) gametocytes on day 11 of development. Gametocytes were cultured as described above in panel **a**. Scale bar = 50 μm. RAPA, rapamycin.

inhibiting _P. falciparum_ parasite multiplication (Supplementary Fig. 9). Finally, we tested the polyhalogenated benzimidazole compound TBB that blocks recombinant human CK2α activity with an $IC_{50}$ of 0.9 μM[61] and DMAT, a structural analogue of TBB that shows improved activity and selectivity towards human CK2α[62]. TBB was previously tested on recombinant PfCK2α and shown to block kinase activity with an $IC_{50}$ of 1.5 μM[12]. In our cell-based assay, TBB showed poor activity ($IC_{50} > 50$ μM) and for DMAT we determined an $IC_{50}$ of 15.8 μM (17.3 μM; 14.4 μM) in inhibiting parasite proliferation (Supplementary Fig. 9).

In summary, we show that four selective inhibitors of human CK2α performed poorly in blocking parasite multiplication in a cell-based assay, despite two of them previously showed promising activity against recombinant PfCK2α in in vitro kinase assays.

## Discussion
In model eukaryotes, the casein kinase CK2 has been implicated in the phosphorylation of a variety of substrates and executes essential cellular functions[17,22–24]. Similarly, in _P. falciparum_ PfCK2 is suggested to phosphorylate a broad range of substrates[10,31,32] and both the regulatory and catalytic kinase subunits are essential for asexual parasite development[10,12,32]. In order to study the function of the catalytic subunit PfCK2α in asexual and sexual development, we generated several transgenic parasites lines using CRISPR/Cas9-based gene editing.

Flow cytometry assays performed using the NF54::DiCre/CK2α-GFP cKO line confirmed the essential role of PfCK2α in merozoite invasion into RBCs[32]. However, whereas the cKO parasite line indeed produced merozoites incapable of invading RBCs, our NF54/AP2-G-mScarlet/CK2α-GFPDD cKD parasite line showed no growth defect upon knocking down PfCK2α expression through Shield-1 removal. This is surprising since Tham and colleagues reported that a very similar cKD line produced in the 3D7 strain (a clonal line of the NF54 strain used in our study), in which the PfCK2α protein is expressed as a 3xHA-DD fusion (PfCK2α-HADD), showed a strong RBC invasion defect[32]. This discrepancy may be explained for instance by higher stability of the PfCK2α-GFPDD protein compared to the PfCK2α-HADD protein in absence of Shield-1, leading to less efficient PfCK2α degradation in our cKD line, or by a slightly higher PfCK2α expression threshold required for successful invasion of 3D7 merozoites. Previous studies have shown that PfCK2 is able to phosphorylate the cytoplasmic tails of both the Rh (Rh2b, Rh4) and EBA (EBA140, EBA175, EBA181) families of invasion ligands using in vitro kinase assays[31,32]. However, direct evidence of lack of phosphorylation of these adhesins in PfCK2-depleted parasites in vivo, for instance by phosphoproteomics studies, is still missing. Interestingly, however, mutational analyses of mapped CK2 phosphosites in the

cytoplasmic tail of Rh4 highlighted their importance in the Rh4-dependent (sialic acid-independent) invasion pathway in vivo[32]. Since 3D7, a clone of the NF54 strain used here, relies on Rh4 for RBC invasion[63], it is likely that the invasion defect in PfCK2α loss-of-function mutants in these strains is indeed due to the lack of PfCK2-dependent phosphorylation of Rh4[63]. Whether PfCK2 is also required for invasion of parasites employing alternative sialic acid-dependent invasion pathways reliant on EBA family ligands is an interesting question to be addressed in the future, for instance by recapitulating PfCK2α loss-of-function mutants in such strains. In this context it is tempting to speculate that the small proportion of RAPA-treated PfCK2α cKO parasites that successfully invaded RBCs may have employed an Rh4-independent invasion pathway potentially dependent on invasion ligand phosphorylation by another kinase[32]. Regardless of this hypothesis, the fact that these parasites failed to progress beyond the trophozoite stage suggests that PfCK2 exhibits essential functions during the IDC in addition to its role in merozoite invasion. This is in line with the proposed pleiotropic activity of CK2 and the variety of putative cellular functions identified for PfCK2 in _P. falciparum_[10]. Interestingly, Dastidar and colleagues revealed a potential function of PfCK2 in chromatin assembly and dynamics, DNA replication and mitosis and could show that PfCK2α is able to phosphorylate histones and nucleosome-assembly proteins in vitro[10]. Lack of post-translational modifications of important chromatin components and regulators such as histones could for instance lead to impaired DNA damage signalling and hence result in a DNA damage-induced cell cycle arrest[30]. Whether PfCK2 indeed phosphorylates histones and other chromatin components in vivo and whether lack of their phosphorylation is also involved in DNA damage sensing and cell cycle arrest in _P. falciparum_ has not yet been investigated[64].

We also studied the role of PfCK2α in gametocytes by analysing both a conditional PfCK2α knockdown line (NF54/AP2-G-mScarlet/CK2α-GFPDD) and a DiCre-inducible PfCK2α knockout line (NF54::DiCre/CK2α-GFP) and demonstrate that PfCK2α is indispensable for gametocytogenesis. To our knowledge, this is the first example of a parasite kinase shown to be essential for sexual development of malaria parasites. The cKD of PfCK2α expression resulted in gametocytes with seemingly normal morphology up to stage IV, but mature crescent-shaped stage V gametocytes with rounded ends, increased cellular deformability and ability to undergo gamete activation and exflagellation were not formed. In comparison to the PfCK2α cKD line, RAPA-induced _pfck2α_ knockout gametocytes showed a more severe phenotype. These gametocytes did not progress past stage II/III of maturation, showed drastic morphological aberrations including lack of elongation and displayed a marked decrease in viability when compared to the DMSO-treated control. Previous research has shown that formation of the inner membrane complex (IMC)

and the tightly associated underlying microtubule and F-actin cytoskeletal networks are main drivers of gametocyte shape and elongation during gametocyte maturation[46,65–68]. The nascent IMC and cytoskeletal structures already appear in stage I/II gametocytes and expand over the whole cell body in stage III/IV gametocytes[46,65–68]. At the stage IV to V transition, the microtubule and actin networks are disassembled and this is linked to the rounding up of both ends of the gametocyte[46,65,66,68] and probably also to their capacity to undergo gamete activation and exflagellation. Interestingly, upon conditional knockdown of two different IMC components (PfPHIL1, PfPIP1) gametocytes failed to develop morphologically beyond stage III, displayed aberrant morphology and a lack of elongation and gametocyte viability decreased drastically over time[66]. Moreover, treatment of gametocytes with the microtubule-destabilising compound trifluralin affects gametocyte morphology[69] and exposure to the actin-stabilising agent jasplakinolide causes marked deformation of late stage gametocytes and a complete lack of stage V formation[65]. The morphologically similar developmental defects observed in our PfCK2α knockdown and knockout gametocytes are therefore indicative for a direct or indirect role of PfCK2 in IMC maturation and/or microtubule or actin cytoskeleton dynamics during gametocytogenesis. While experimental evidence for PfCK2-dependent phosphorylation of IMC components is currently lacking, CK2 has been shown to interact directly with tubulin and microtubules in human cells and several *Trypanosoma* species[70–73] and to phosphorylate microtubule-associated proteins[74]. In addition, human CK2 is directly involved in the regulation of microtubule assembly and stability[75] and inhibition of CK2 activity induces alterations to the cytoskeleton and shape of astrocytes and endothelial cells[76]. Moreover, in the apicomplexan parasite *Toxoplasma gondii* phosphorylation of the actin-binding protein toxofilin by a CK2-like activity was shown to be important for the actin-toxofilin interaction and consequently actin dynamics[77].

In contrast to day 11 control gametocytes, PfCK2α knockdown gametocytes retained a high level of cellular rigidity similar to immature stage III gametocytes. While the elaborate IMC and cytoskeletal network structures in the parasite may contribute to the increased rigidity of immature gametocytes, parasite-induced remodelling of the membrane and underlying spectrin and actin networks of the host RBC seems to play a more dominant role in gametocyte rigidification[43,69,78–81]. The deformability switch occurring at the stage IV to V transition is linked to the dephosphorylation and dissociation of the parasite-encoded STEVOR protein from the iRBC membrane[43,79,81] and to the reversal of previously established cytoskeletal rearrangements underneath[69]. Our findings therefore indicate that PfCK2α-depleted stage IV gametocytes are either arrested in development prior to the initiation of the deformability switch or that PfCK2-dependent signalling is required to regulate this process. Given that PfCK2 is not known to be exported into the RBC cytosol, however, the direct phosphorylation of proteins in the iRBC by PfCK2 seems rather unlikely. In summary, our results show that PfCK2 is essential for the morphological and functional maturation of gametocytes and provide circumstantial evidence that PfCK2 may be required for the regulation of IMC biogenesis and cytoskeleton dynamics in the parasite and/or host RBC. Alternatively, it is also conceivable that PfCK2 may be important for the proper control of gene expression during gametocytogenesis. PfCK2 shows prominent nuclear localisation in gametocytes, interacts with chromatin components in asexual parasites[10] and CK2 is known to regulate the activity of transcriptional regulators in yeast[20,21]. Notwithstanding the above hypotheses, the findings and transgenic parasite lines generated in this study will allow for a detailed targeted analysis of PfCK2 function in gametocytes using complementary biochemical, cell biological and high throughput approaches.

The essential roles PfCK2α plays in asexual and sexual development of *P. falciparum* renders this kinase an even more attractive drug target than anticipated previously. A drug specifically targeting *P. falciparum* PfCK2α might therefore combine asexual parasite clearance and transmission-blocking properties. All CK2 kinase-specific motifs have been identified in PfCK2α protein and it shares 65% amino acid sequence identity with its human CK2α orthologue (HsCK2α)[12,25,26]. Despite the high similarity of PfCK2α and HsCK2α, Holland and colleagues (2009) showed by using the CK2 inhibitor Rottlerin that selective inhibition of PfCK2α is achievable[12]. In our study, we tested Quinalizarin. TTP 22, TBB and DMAT for their ability to inhibit parasite multiplication. The $IC_{50}$ concentrations of DMAT and the other three CK2α inhibitors were found to be more than 1,000-fold higher (15.8 μM; >50 μM; >50 μM; >50 μM) when compared with the antimalarial control drugs Chloroquine and Artesunate (9.1 nM; 6.8 nM). Several previous studies have shown inhibition of recombinant or purified PfCK2α at low $IC_{50}$ values[11–13]. Unfortunately, however, it becomes evident that these promising results from in vitro kinase assays do not translate into activity of the tested compounds against parasite growth and multiplication. Nevertheless, screening for *Plasmodium*-specific and membrane-permeable PfCK2α inhibitors and lead optimisation may still be worthwhile pursuing, given that PfCK2α is essential for both asexual proliferation and gametocytogenesis. Of note, several CK2 inhibitors have been produced or are under development for use in cancer treatment[82–84] and compounds stemming from these campaigns may be a promising starting point to discover compounds targeting PfCK2α in vivo.

## Methods

**Parasite culture**. Culturing and sorbitol synchronization of *P. falciparum* NF54 parasites[85] were performed as described[86,87]. For routine parasite culture, parasites were propagated in AB + or B + human RBCs (Blood Donation Center, Zurich, Switzerland) at a hematocrit of 5% in complete parasite culture medium (PCM) consisting of 10.44 g/liter RPMI-1640, 25 mM HEPES, 100 μM hypoxanthine, 24 mM sodium bicarbonate and 0.5% AlbuMAX II (Gibco #11021-037). The medium was further complemented with 2 mM choline chloride (Sigma #C7527) to suppress sexual commitment[37], Parasite cultures were incubated at 37 °C in air-tight containers filled with malaria gas (4% $CO_2$, 3% $O_2$, 93% $N_2$).

**Cloning of transfection constructs**. CRISPR/Cas9-based genome engineering of parasites was performed using a two-plasmid approach as previously described[38]. This system is based on co-transfection of a suicide and donor plasmid. The suicide plasmid contains expression cassettes for the Cas9 enzyme, the single guide RNA (sgRNA) and either the human dihydrofolate reductase (hDHFR) or blasticidin deaminase (BSD) resistance markers (pH-gC, pB-gC). In two additional suicide plasmids the resistance marker hDHFR or BSD is further fused to the negative selection marker yeast cytosine deaminase/uridyl phosphoribosyl transferase (yFCU) (pHF-gC or pBF-gC). A pD-derived donor plasmid is needed for homology-directed repair of the Cas9-induced DNA lesion[38]. For this study we generated an additional suicide plasmid, pY-gC, containing the yeast dihydroorotate dehydrogenase (yDHODH) resistance cassette conferring resistance to DSM1[88] instead of the h*dhfr(-fcu)* or *bsd(-fcu)* cassettes. The y*dhodh* resistance cassette was amplified from the pUF1-Cas9 plasmid[89] by producing two overlapping PCR fragments (F1 and F2) in order to mutate the *BsaI* restriction enzyme recognition site present in the y*dhodh* coding sequence. To this end, we inserted a single point mutation in the complementary reverse and forward primers used to amplify the F1 and F2 fragments, respectively. F1 was amplified using primers Y_F and B_R and F2 using primers B_F and Y_R. The pY-gC plasmid was then generated in a three-fragment Gibson assembly joining the *BamHI* and *XhoI* fragment of the pBF-gC plasmid[38] and the two overlapping y*dhodh* fragments F1 and F2. The resulting pY-gC plasmid and the previously published suicide plasmids pHF-gC and pBF-gC[38] were used for insertion of sgRNA-encoding sequences targeting the 3′ end (sgt_*ck2a_1*) or 5′ end (sgt_*ck2a_2*) of *pfckα*. For this purpose, complementary oligonucleotides were annealed and the resulting double-stranded fragments were ligated into the *BsaI* digested pY-gC, pBF-gC or pHF-gC plasmids

using T4 DNA ligase generating the pY-gC_ck2α_tag, pBF-gC_ck2α_tag, and pHF-gC_ck2α-cKO plasmids.

Plasmids pBF-gC_ck2a_tag and pY-gC_ck2a_tag, both encoding the sgt_ck2a_1 sgRNA, were used to generate parasites expressing PfCK2α-GFP and PfCK2α-GFPDD, respectively. The corresponding donor plasmids pD_ck2α-gfp and pD_ck2α-gfpdd were produced by assembling four PCR fragments in a Gibson reaction using (a) the plasmid backbone amplified from pUC19 using primers PCRA_F and PCRA_R[38], (b) the 5′ homology region (HR) amplified from NF54 gDNA using primers H1_F and H1_R, (c) the gfp (primers G_F and G_R) or gfpdd (primers G_F and GD_R) sequence amplified from pHcamGDV1-GFP-DD[38], and (d) the 3′ HR amplified from NF54 gDNA using primers H2_F and H2_R (pD_ck2α-gfp) or H2D_F and H2_R (pD_ck2α-gfpdd).

The NF54::DiCre/CK2α_cKO parasite line was generated in two consecutive transfection steps using the suicide plasmids pBF-gC_ck2α_tag (encoding the sgt_ck2a_1 sgRNA) and pHF-gC_ck2α-cKO (encoding the sgt_ck2a_2 sgRNA) and the corresponding donor plasmids pD_ck2α-cKO1 and pD_ck2α-cKO2. The pD_ck2α-cKO1 plasmid was generated in a three-fragment Gibson reaction joining (a) the plasmid backbone amplified from pUC19 using primers PCRA_F and PCRA_R, as previously described[38], (b) the 5′ HR followed by the in-frame gfp coding sequence amplified from pD_ck2α-gfp using primers H1_F and loxP1_R, and (c) the 3′ HR amplified from pD_ck2α-gfp using primers loxP1_F and H2_R. The loxP sequence[54] downstream of the gfp coding sequence was introduced into the primers loxP1_F and loxP1_R used to amplify the corresponding PCR fragments. The second donor plasmid pD_ck2α-cKO2 was generated by assembling four fragments in a Gibson reaction using (a) the plasmid backbone amplified from pUC19 using primers PCRA_F and PCRA_R[38], (b) the 5′ HR amplified from NF54 gDNA using primers H3_F and H3_R, (c) the sera2 intron:loxP fragment (loxPint)[53] amplified from the pD_SIP2xGFP plasmid (I. Niederwieser, unpublished) using primers loxPINT_F and loxPINT_R, and (d) the 3′ HR amplified from NF54 gDNA using primers H4_F and H4_R. All oligonucleotides used for the cloning of transfection constructs and annealing of sgRNA-encoding sequences are listed in Supplementary Table 1.

**Transfection and transgenic cell lines**. P. falciparum transfection using the CRISPR/Cas9 suicide and donor constructs was performed as explained previously[38]. All transgenic cell lines were generated by transfecting parasites with 50 μg each of the suicide plasmid and the respective donor plasmid. Depending on the suicide plasmid used (pBF-gC_ck2α_tag, pHF-gC_ck2α-cKO, pY-gC_ck2α_tag) the transfected populations were treated either with 2.5 μg/mL blasticidin-S-hydrochloride (for 10 subsequent days), 4 nM WR99210 (for six subsequent days) or 1.5 μM DSM1 (for 10 subsequent days). The plasmid combinations pBF-gC_ck2_tag/pD_ck2α-gfp and pY-gC_ck2_tag/pD_ck2α-gfpdd were transfected into the NF54/AP2-G-mScarlet line to generate the NF54/AP2-G-mScarlet/CK2α-GFP and NF54/AP2-G-mScarlet/CK2α-GFPDD lines, respectively. The NF54/AP2-G-mScarlet line has previously been engineered to express the PfAP2-G transcription factor as a C-terminal mScarlet fusion (Brancucci et al., manuscript in preparation). After transfection, NF54/AP2-G-mScarlet/CK2α-GFPDD parasites were constantly cultured on 625 nM Shield-1 to stabilize expressed PfCK2α-GFPDD proteins. The NF54::DiCre parasite line[52] (kind gift from Moritz Treeck) was transfected first with the pBF-gC_ck2α_tag suicide and pD_ck2α-cKO1 donor plasmids. After successful selection of the transgenic line a second transfection step using the pHF-gC_ck2α-cKO suicide and pD_ck2α-cKO2 donor plasmids was performed to obtain the NF54::DiCre/CK2α_cKO parasite line carrying loxP elements at the 5′ and 3′ end of the pfck2α-gfp gene. Transgenic populations were routinely obtained 2–3 weeks after transfection and correct editing of the engineered loci was confirmed by PCR on gDNA. Because editing of the targeted locus was 100% efficient in all transgenic lines based on the diagnostic PCR results, the lines were not cloned out prior to further investigation. Schematic maps of the transfection plasmids, transgenic loci, PCR primer binding sites and results of the diagnostic PCRs can be found in Supplementary Figs. 1, 2 and 6. Sequences of all primers used for diagnostic PCR reactions are listed in Supplementary Table 2.

**Fluorescence microscopy**. For live cell fluorescence microscopy, parasite DNA was stained using 5 μg/ml Hoechst (Merck) and the microscopy slides were mounted using Vectashield (Vector Laboratories). Slides were viewed using the Leica DM5000 B fluorescence microscope (20×, 40×, and 63× objectives and 10× or 16× microscope magnification) and images were acquired using the Leica DFC345 FX camera and the Leica application suite software (LAS 4.9.0). Images were processed using Adobe Photoshop CC 2018. In each experiment, identical settings were used for both image acquisition and processing.

**Western blot analysis**. Parasites were released from iRBCs using 0.15% saponin in PBS and incubation on ice for 10 min. Parasite pellets were washed in ice-cold PBS 2–3 times until the supernatant was clear. An Urea/SDS lysis buffer (8 M Urea, 5% SDS, 50 mM Bis-Tris, 2 mM EDTA, 25 mM HCl, pH 6.5) complemented with 1x protease inhibitor cocktail (Merck) and 1 mM DTT was used to obtain whole cell protein lysates. Nuclear and cytoplasmic lysates were obtained by first lysing parasite pellets in cytoplasmic lysis buffer (20 mM Hepes, 10 mM KCl, 1 mM

EDTA, 0.65% Igepal) complemented with 1x protease inhibitor cocktail (Merck) and 1 mM DTT. The resulting supernatant containing the cytoplasmic protein fraction was collected. The nuclear pellet was washed in cytoplasmic lysis buffer until the supernatant was clear followed by lysis of the nuclei in Urea/SDS buffer. Protein lysates were separated either on NuPage 3–8% Tris-Acetate or 5–12% Bis-Tris gels (Novex, Qiagen) using MES running buffer (Novex, Qiagen). Upon protein transfer, the nitrocellulose membrane was blocked in 5% milk in PBS/0.1% Tween (PBS-Tween) for 30 min. Protein detection was performed using mouse mAb α-GFP (1:1,000) (Roche Diagnostics #11814460001), mAb α-PfGAPDH (1:20,000)[90] and rabbit α-PfHP1 (1:5,000)[91] diluted in blocking buffer. Incubation with the first antibody was performed at 4 °C over night. After 3-4 washing steps in PBS/Tween, the membrane was incubated for at least 1 h using the secondary antibody goat α-mouse IgG (H&L)-HRP (1:10,000) (GE healthcare #NXA931) or donkey anti-rabbit IgG (H&L) HRP (1:5,000) (GE Healthcare #NA934) diluted in blocking buffer. Membranes were subsequently washed 3–4 times using PBS/Tween before signal detection. The α-GFP/α-GAPDH blot shown in Supplementary Fig. 3 was stripped in 2% SDS, 62.5 mM Tris-HCl (pH 6.8), 100 mM β-mercaptoethanol for 30 min at 60 °C and re-probed with rabbit α-PfHP1 antibodies[91].

**Quantification of sexual commitment rates**. NF54/AP2-G-mScarlet/CK2α-GFPDD parasites were synchronised to a 6-h time window by two consecutive treatments with 5% sorbitol. In the following replication cycle, parasite cultures were exposed to ±Shield and either −SerM (sexual commitment inducing) or −SerM/CC (sexual commitment inhibiting) conditions at 18–24 hpi (2% parasitaemia, 2.5% hematocrit)[37]. After a 48-hour incubation period, cultures were resuspended and 30 μL of this suspension was mixed with 50 μL PBS containing 8.1 μM DNA dye Hoechst 33342. Cells were incubated in a 96 well plate for 30 min, pelleted at 300 g for 5 min, washed twice using 200 μL PBS, and resuspended in 180 μL PBS. A total of 30 μL of this cell suspension were mixed with 150 μL PBS within wells of a clear-bottom 96 well plate (Greiner CELLCOAT microplate 655948, Poly-D-Lysine, flat μClear bottom) and cells were allowed to settle for 30 min prior to imaging. Images were acquired using an ImageXpress Micro widefield high content screening system (Molecular Devices) in combination with MetaXpress software (version 6.5.4.532, Molecular Devices) and a Sola SE solid state white light engine (Lumencor). Filtersets for Hoechst (Ex: 377/50 nm, Em: 447/60 nm) and mScarlet (Ex: 543/22 nm, Em: 593/40 nm) were used with exposure times of 80 ms and 600 ms, respectively. 36 sites per well were imaged using a Plan-Apochromat 40x objective. Images were analysed using the MetaXpress software and Hoechst-positive as well as mScarlet-positive parasites were quantified allowing for the calculation of sexual commitment rates corresponding to the percentage of mScarlet-positive parasites amongst all parasites (Hoechst-positive).

**Gametocyte cultures**. Synchronous gametocyte populations were needed for morphological assessment, extraction of protein samples, microsphiltration experiments and exflagellation assays. For this purpose, sexual commitment was induced by culturing trophozoites in −SerM medium for 24 h, as described above[37]. After reinvasion (0–6 hpi) (asexual/sexual ring stages), parasites were cultured in PCM containing 10% human serum instead of 0.5% AlbuMAX II (+SerM). At 24–30 hpi (trophozoites and stage I gametocytes), 50 mM N-acetyl-D-glucosamine (Sigma) was added to the +SerM medium (+SerM/GlcNAc) for six consecutive days to eliminate asexual parasites[42]. From day seven onwards, gametocytes were cultured in +SerM and daily medium changes were performed on a 37 °C heating plate.

**Exflagellation assays**. On day 14 of gametocyte maturation, stage V gametocytes were subjected to exflagellation assays[50]. In brief, gametocytes were activated using a temperature drop (from 37 °C to 22 °C/room temperature) and 100 μM xanthurenic acid (XA) for 15 min. Subsequently, the number of exflagellation centres and total RBCs per mL of culture were determined by bright-field microscopy using a Neubauer chamber. Gametocytaemia was determined by visual inspection of Giemsa-stained blood smears before exflagellation thus allowing to calculate exflagellation rates as the number of exflagellating parasites per total number of gametocytes.

**Microsphiltration experiments**. Synchronous NF54/AP2-G-mScarlet/CK2α-GFPDD parasites were split at 0–6 hpi and cultured either in the presence of 675 nM Shield-1 (+Shield-1) or in absence of Shield-1 (−Shield-1). Induction of sexual commitment using −SerM medium and culturing of gametocytes using +SerM and +SerM/GlcNAc media was performed as described above. Microsphiltration experiments were conducted in two independent biological replicates as published previously[44] on day seven (stage III) and day 11 (stage V) of gametocyte development. In detail, aliquots of the cultures were transferred to 15 mL tubes and the hematocrit was lowered to 1.5% by addition of fresh PCM. Tubes were kept in a water bath at 37 °C to inhibit rounding up and exflagellation of mature stage V gametocytes. For each condition (±Shield-1), six microsphiltration columns (technical replicates) were loaded. Per column, 600 μL of cell suspension was injected and washed through the column with 5 mL +SerM medium at 60 mL/

h using a medical-grade pump (Syramed μSP6000, Acromed AG, Switzerland). To determine gametocyte retention rates, Giemsa-stained thin smears were prepared before and after the samples were subjected to microspiltration. The input gametocytaemia before passing the sample through the column ("UP" gametocytaemia) was determined as the average count from two Giemsa-stained blood smears. The gametocytaemia in the microspiltration elution fractions ("DOWN" gametocytaemia) was determined from Giemsa-stained thin smears prepared from each individual elution tube. At least 1000 total RBCs were counted per slide. Gametocyte retention rates were calculated as 1-("DOWN" gametocytaemia/"UP" gametocytaemia). The blood smears prepared from the day 11 input samples additionally served as quality control to confirm that stage V gametocytes did not undergo rounding up and exflagellation.

**Flow cytometry analysis**. For parasite multiplication assays, synchronous NF54/AP2-G-mScarlet/CK2α-GFPDD parasites were split at 0–6 hpi (0.2% parasitaemia) and cultured either in presence of 675 nM Shield-1 (+Shield-1) or in absence of Shield-1 (−Shield-1) during the whole duration of the assay. Synchronous NF54::DiCre/CK2α-GFP cKO parasites were split at 0–6 hpi (0.2% parasitaemia) and exposed for 4 h to 100 nM RAPA to trigger excision of the *pfck2α-gfp* gene (DMSO was added to the control population instead of RAPA)[52,54]. Subsequently, the RBCs were washed once in PCM/2 mM choline chloride and resuspended in this medium for onward in vitro culture. After 18 h (18–24 hpi) parasite DNA was stained at 37 °C for 30 min using SYBR Green DNA stain (1:10,000, Invitrogen) to determine the starting parasitaemia (day 1). Flow cytometry measurements were repeated on day 3 (generation 2) and day 5 (generation 3) for NF54/AP2-G-mScarlet/CK2α-GFPDD parasites (±Shield-1) and on day 3 (generation 2), day 5 (generation 3), and day 7 (generation 4) for NF54::DiCre/CK2α-GFP cKO parasites (RAPA/DMSO).

To analyse parasite progression through schizogony and merozoite egress, synchronized NF54::DiCre/CK2α-GFP cKO parasites (0–4 hpi) were treated with RAPA or DMSO as explained above. Samples were collected for DNA content analysis starting at 20–24 hpi up to 20–24 hpi in the following generation. For merozoite egress experiments, 1 μM compound 2 (C2) was added to the cultures from 36 to 40 hpi onwards to prevent schizont rupture[55]. At 50–54 hpi to 54–58 hpi, C2-arrested segmented schizont cultures were split and one half was directly inactivated by fixation in 4% formaldehyde/0.0075% glutaraldehyde (C2-arrested control). The other half of the sample was washed once in culture medium to remove C2, resuspended in culture medium, and rotated at 37 °C to allow merozoite egress and invasion for 45 min (replicate 1) and 45 min and 90 min (replicate 2) before samples were fixed in 4% formaldehyde/0.0075% glutaraldehyde. Fixed samples were washed twice in PBS and permeabilized for 15 min in PBS containing 0.1%Triton X-100 and 0.1 mg/ml RNase A. Fixed and permeabilized cells were washed twice in PBS and stained with SYBR Green DNA stain (1:5,000, Invitrogen) for 30 min.

Fluorescence intensities were measured using the MACS Quant Analyzer 10 (200,000 RBCs measured per sample). Data were analysed using the FlowJo_v10.6.1 software. Gating was performed to remove small debris (smaller than cell size), doublets (single measurement event consisting of two cells), and to separate uninfected from infected RBCs (using an uninfected RBC control sample) based on their SYBR green intensity (Supplementary Fig. 4). For the results presented in Supplementary Fig. 7b an additional gate was set to separate mature schizonts from remaining iRBCs based on SYBR green intensity (Supplementary Fig. 8).

**Drug assays**. Activity of CK2 kinase inhibitors on asexual NF54 parasite multiplication was determined using an [³H] hypoxanthine incorporation assay[58]. The mean $IC_{50}$ values were determined from two biological replicate assays. One of the biological replicates was performed in two technical replicates. Quinalizarin (CAS Number 81-61-8, Sigma #Q2763), TBB (CAS Number 17374-26-4, Selleckchem #S5265), DMAT (CAS Number 749234-11-5, Sigma #SML2044) and TTP 22 (CAS Number 329907-28-0, TOCRIS #4432), were resuspended in DMSO and used at a maximum starting concentration of 50 μM. In a six-step serial dilution, the compound concentration was diluted to half in each step to span a concentration range between 50 μM and 0.8 μM. Chloroquine (CAS Number 50-63-5, Sigma #C6628) and Artesunate (CAS Number 88495-63-0, Mepha #11665) served as control antimalarial compounds with a starting concentration of 193 nM and 26 nM, respectively. In a six-step serial dilution, the Chloroquine and Artesunate concentrations were diluted to half in each step to span a concentration range between 193 nM and 3 nM or between 26 nM and 0.4 nM, respectively.

**Statistics and reproducibility**. All data from assays quantifying parasite multiplication, sexual commitment rates, morphological gametocyte staging, gametocyte deformability, or exflagellation rates were represented as means with error bars defining the standard deviation. All data were derived from at least three biological replicate experiments. Statistical significance was determined using paired or unpaired Student's *t* test as indicated in the figure legends. The exact number of biological replicates performed per experiment and the number of cells analysed per sample are indicated in the figure legends and in Supplementary Data 1. Data were analysed and plotted using RStudio Version 1.1.456 and package ggplot2.

**Reporting summary**. Further information on research design is available in the Nature Research Reporting Summary linked to this article.

## Data availability

The source data for all the graphs and charts in the main figures are present in Supplementary Data 1 and any other remaining information can be available from the corresponding author upon reasonable request.

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

## Acknowledgements

We thank M. Treeck for providing the NF54::DiCre line. This work was supported by funding from the Swiss National Science Foundation (BSCGI0_157729, 31003A_169347, 310030_184785) and the Rudolf Geigy Foundation.

## Author contributions

E.H. and O.G. generated all transgenic parasite lines, designed and performed experiments, analysed, and interpreted data. E.H. prepared illustrations and wrote the manuscript. A.P. designed, performed, and analysed microsphiltration experiments and NF54/AP2-G-mScarlet/CK2α-GFPDD gametocyte maturation assays. M. W. performed and analysed NF54::DiCre/CK2α-GFP cKO flow cytometry experiments. N.M.B.B and H.P.B helped designing and supervised experiments and provided resources. C.S. helped designing, performing and analysing drug assays and S.W. supervised these experiments and provided resources. T.S.V. conceived of the study, designed and supervised experiments, provided resources, and wrote the manuscript. All authors contributed to the editing of the manuscript.

## Competing interests

The authors declare no competing interests.
