## [Peer Review File · Communications Biology]

Reviewers' comments:

Reviewer #1 (Remarks to the Author):

Summary

The effect of knock and down or gene deletion of PfCK2a is investigated in both asexual and sexual stages of *P. falciparum*. In asexuals they find a significant difference in the phenotype of PfCK2a KD parasites when compared with previous work by Tham et al. To further investigate this difference a cKO line is generated which more closely aligns with the previous findings. The role of PfCK2a in gametocytogenesis is also investigated. The KD line results in morphological defects in late gametocyte development that prevent increased deformability of Stage V gametocytes. Late stage KD gams are also less responsive to activation stimuli. They then show with the cKO line that gametocyte development is drastically impaired by excision of CK2a. They test inhibitors of CK2a on parasites and find them to be poorly inhibitory to parasite growth. The work that has been done is to a high standard. I find that some of the phenotypes don't tightly correspond to the abundance of CK2a by Western which I think could be investigated further and discussed in greater depth. As it stands the work shows an important role across both sexual and asexual blood stages for PfCK2a. The mechanistic reason for the defects is not determined but well-reasoned possibilities discussed.

Minor comments

1. Just for clarity, parasites weren't cloned, and therefore your lines represent a mixed population of desired integration alongside various plasmid concatemer (misspelt in the fig legend) integrations?
2. "Such undesired recombination events have previously been reported in other studies as well 38,39. However, since the 829 bp 3' homology region (HR) used for homology-directed repair includes the native terminator 40, expression of the modified pfck2a genes is not compromised by donor plasmid integration (Figs. 1 and 2 and Supplementary Figs. 1-3). ". Without comparison of clonal lines, clean vs concatemer, it can't be definitively stated that expression is not compromised. I doubt it is, but without direct evidence this needs to be less definitive in the text.
3. In light of the completely different phenotype between Tham and this work with a very similar parasite I think the relative abundance of protein needs further investigation. Could they perform a timecourse with/out shield and reproduce fig 3b from Tham et al, also providing densitometry? From the Western in Supp 2b their KD of PfCK2a looks at least as strong as that of Tham, and given the similarity in parasite strains used 3D7 and NF54, it would be nice to drill down on this difference in results a little more. Relevant to this line in discussion: "This discrepancy may be explained by higher stability of the PfCK2a-GFPDD protein compared to the PfCK2a-HADD protein in absence of Shield-1, leading to less efficient PfCK2a degradation in our cKD line."
4. Typo: Gametocyte activation can be observed by a change in shape since gametocytes egress from the iRBC and become spherical in a process termed "rounding up" 48
5. With some shuffling there is a plenty of room in figure 3 and I find the data in supp fig 5 interesting. I suggest these data are all included in the primary manuscript to save the reader so much chopping around.
6. Gametocyte experiments can be quite variable. Could the experiments in supp 5 be repeated to confirm the findings, currently stated as one replicate.
7. It would also be nice to have this finding "Whereas the morphology of stage I-IV gametocytes (day 2 to 9) was comparable between NF54/AP2-G-mScarlet/CK2a-GFPDD parasites cultured in presence or absence of Shield-1, most PfCK2a-GFPDD-depleted gametocytes failed to develop into mature stage V gametocytes (day 11), even after prolonged periods of observation " supported with a figure similar to that in Supp 7d.
8. Western blot in Supp 7b. The loading controls are markedly different due to the growth defect in RAPA treatment parasites. Could they provide relative abundance against loading control to determine the quantity of PfCK2a that's here? It's markedly higher than in the in the asexual cKO.

Also, when compared to Supp 5a there is more GFP on the blot for cKO than for KD yet the phenotype is so much stronger in the KO. Where is all the protein coming from in the cKO gams? If the protein is lingering, then why isn't it functional? The small amount present in the KD parasites seems adequate to pull them through to Stage IV before they become seriously defective. The reason given for the difference between Tham and this work is more efficient degradation of protein in the 3HADD parasite, but then within this work there are 2 experiments investigating gametocytes and it looks like the stronger reduction in protein level leads to a weaker phenotype. Alongside the different findings between Tham et al and this work, I think this warrants further investigation and discussion.

9. Again, I feel that most of figure Supp7 should appear in the main manuscript. At least c and d but really the whole lot. I personally find this supp figure more interesting than the main Fig 5.

10. It would be nice to also include a brief description/background of DMAT/TBB as you have done for the other inhibitors. Eg "TTP 22, a cell-permeable ATP analog shown to inhibit the human CK2 kinase in vitro"

11. "In summary, we show that three inhibitors of human CK2 α of which two showed promising activity against recombinant PfCK2 α in in vitro kinase assays performed poorly in blocking parasite multiplication in a cell-based assay."

I had to reread this to understand it. Perhaps like this?:

In summary, we show that three inhibitors of human CK2 α performed poorly in blocking parasite multiplication in a cell-based assay, despite two of these having previously shown promising activity against recombinant PfCK2 α in in vitro kinase assays.

a. Also, this is inaccurate. You state TBB was previously tested on PfCK2 α , you've used DMAT, therefore only one of these compounds has truly been tested on PfCK2 α . What was the reason for not using TBB (especially given there are existing IC50 data)? DMAT has improved activity against human CK2 α , this may not carry over to PfCK2 α . Given that you have some activity in DMAT and a structural analog TBB, I think this would be interesting to also test.

Reviewer #2 (Remarks to the Author):

This manuscript describes the role of protein kinase PfCK2 α in the development of human malaria parasite *P. falciparum*. Since this kinase is essential for the blood stage development of the parasite, two different approaches were taken to generate a condition knockout/knockdown to investigate its function. Using mutant parasites, role of PfCK2 α was established in asexual as well as sexual development of the parasite. The role of this kinase in asexual development and possibly in invasion has previously been reported (Tham et al. *Plos Pathogens* 2015). Therefore, the major novel finding in this study is its involvement in gametocyte development. The following issues need to be addressed:

1. The FKBP-DD based approach to knockdown PfCK2 α did not alter the asexual development. Using a similar approach, Tham et al., found major defect in parasite growth/invasion. How can these differences be explained? Could it be the different background strain used in the two studies? This and other possibilities should be discussed. Given that there is no phenotype in asexual stages and only a modest difference in late stage gametocyte difference, the data related to FKBP-GFP-DD should be removed or moved to the supplement.

2. The KO using diCre/loxP resulted in major defects in parasitemia in second and subsequent cycle and authors have attributed this to defects in invasion. However, they have not performed direct assays to assess invasion and egress. It is also important to quantify various intraerythrocytic developmental stages to rule out the defects on IDC. In addition, the number of merozoites formed should also be determined, which is important to assess the role in parasite replication especially when this kinase is present in the nucleus.

3. Did CK2 inhibitors block gametocytogenesis? The efficacy of these inhibitors should also be tested after PfCK2 α depletion.

Author response to reviewers' comments on "The catalytic subunit of Plasmodium falciparum casein kinase 2 is essential for gametocytogenesis" by Hitz et al.

We thank both reviewers for their fair and critical assessment of our manuscript. Please find our point-by-point responses below.

Reviewer #1 (Remarks to the Author)

The effect of knock and down or gene deletion of PfCK2a is investigated in both asexual and sexual stages of *P. falciparum*. In asexuals they find a significant difference in the phenotype of PfCK2a KD parasites when compared with previous work by Tham et al. To further investigate this difference a cKO line is generated which more closely aligns with the previous findings.

The role of PfCK2a in gametocytogenesis is also investigated. The KD line results in morphological defects in late gametocyte development that prevent increased deformability of Stage V gametocytes. Late stage KD gams are also less responsive to activation stimuli. They then show with the cKO line that gametocyte development is drastically impaired by excision of CK2a. They test inhibitors of CK2a on parasites and find them to be poorly inhibitory to parasite growth.

The work that has been done is to a high standard. I find that some of the phenotypes don't tightly correspond to the abundance of CK2a by Western which I think could be investigated further and discussed in greater depth. As it stands the work shows an important role across both sexual and asexual blood stages for PfCK2a. The mechanistic reason for the defects is not determined but well-reasoned possibilities discussed.

Minor comments

1. Just for clarity, parasites weren't cloned, and therefore your lines represent a mixed population of desired integration alongside various plasmid concatemer (misspelt in the fig legend) integrations?

Yes this is correct. We now mention that we didn't clone out the transgenic lines in the Methods section (line 519f.):

"Because editing of the targeted locus was 100% efficient in all transgenic lines based on the diagnostic PCR results, the lines were not cloned out prior to further investigation."

2. "Such undesired recombination events have previously been reported in other studies as well 38,39. However, since the 829 bp 3' homology region (HR) used for homology-directed repair includes the native terminator 40, expression of the modified *pfck2* genes is not compromised by donor plasmid integration (Figs. 1 and 2 and Supplementary Figs. 1-3). ". Without comparison of clonal lines, clean vs concatemer, it can't be definitively stated that expression is not compromised. I doubt it is, but without direct evidence this needs to be less definitive in the text.

We agree with reviewer 1 and changed this sentence accordingly (line 117ff.):

"However, since the 829 bp 3' homology region (HR) used for homology-directed repair ~~includes~~seems to include the native terminator based on published RNA-seq data⁴⁰, expression of the modified *pfck2a* genes is likely not compromised by donor plasmid integration (Figs. 1 and 2 and Supplementary Figs. 1-3)."

3. In light of the completely different phenotype between Tham and this work with a very similar parasite I think the relative abundance of protein needs further investigation. Could they perform a timecourse with/out shield and reproduce fig 3b from Tham et al, also providing densitometry? From the Western in Supp 2b their KD of PfCK2a looks at least as strong as that of Tham, and given the similarity in parasite strains used 3D7 and NF54, it would be nice to drill down on this difference in results a little more. Relevant to this line in discussion: "This discrepancy may be explained by higher stability of the PfCK2 -GFPDD protein compared to

the PfCK2 -HADD protein in absence of Shield-1, leading to less efficient PfCK2 degradation in our cKD line.“

We would like to stress that the PfCK2 loss-of-function phenotype was not completely different but rather identical in both studies. Tham et al. found that the Shield-1-inducible “knockdown” of their HA-DD-tagged PfCK2 enzyme was lethal and that PfCK2 is essential for invasion in 3D7 parasites. Likewise, we show that the rapamycin-inducible “knockout” of the *pfck2α* gene is lethal and that PfCK2 is essential for invasion in NF54 parasites (NF54::DiCre/CK2_cKO) (previous and new data; please see point 2 from reviewer 2 below). The fact that the Shield-1-inducible “knockdown” of GFP-DD-tagged PfCK2 in our NF54/CK2 -GFPDD KD line was not lethal simply means that the residual PfCK2 -GFPDD expression levels in these NF54 parasites are still sufficient to allow invasion. Quantifying the knockdown efficiency of PfCK2 -GFPDD expression by WB and densitometry will not change this conclusion. We therefore don't see an added value of performing a WB time course experiment with our cell line. Furthermore, we don't think a comparison with the densitometry data from a single time course WB experiment from the Tham et al. study would provide meaningful results in the first place (this would require triplicate experiments in both studies). Moreover, the WB results in Fig. 3b from Tham et al. are not comparable to our WB results shown in Fig. S2b since Tham et al. analysed protein samples harvested during the first 12 hours of Shield-1 removal, while our samples were harvested 40 hours after Shield-1 removal. Whether PfCK2 -GFPDD (our study) has higher stability in absence of Shield-1 compared to PfCK2 -HADD (Tham et al. study) or whether 3D7 parasites (Tham et al. study) require higher PfCK2 expression levels for successful invasion compared to NF54 parasites (our study) can only be addressed by tagging PfCK2 with GFPDD in the 3D7 strain used by Tham et al. and with HADD in the NF54 strain used by us, and comparing all four transgenic lines simultaneously. We believe performing these experiments is out of scope of our study as we clearly demonstrate with the NF54::DiCre/CK2_cKO line that PfCK2 is essential. We therefore propose to slightly alter the sentence reviewer 1 is referring to (line 339ff.):

"This discrepancy may be explained for instance by higher stability of the PfCK2α-GFPDD protein compared to the PfCK2α-HADD protein in absence of Shield-1, leading to less efficient PfCK2α degradation in our cKD line, or by a slightly higher PfCK2α expression threshold required for successful invasion of 3D7 merozoites."

4. Typo: Gametocyte activation can be observed by a change in shape since gametocytes egress from the iRBC and become spherical in a process termed “rounding up” 48.

We corrected this typo (line 185):

"formfrom"

5. With some shuffling there is a plenty of room in figure 3 and I find the data in supp fig 5 interesting. I suggest these data are all included in the primary manuscript to save the reader so much chopping around.

We revised Fig. 3 such that it now combines most of the data from the original Fig. 3 and Fig. S5. The revised Fig. S5 now only shows the sexual conversion rate data (original Fig. 3, panel a) and the full-length WBs.

Revised Figure 3:

Revised Figure S5:

6. Gametocyte experiments can be quite variable. Could the experiments in supp 5 be repeated to confirm the findings, currently stated as one replicate.

We repeated this experiment in three additional biological replicates and the data are shown in Fig. 3f (please see revised Figure 3 above).

7. It would also be nice to have this finding “Whereas the morphology of stage I-IV gametocytes (day 2 to 9) was comparable between NF54/AP2-G-mScarlet/CK2 -GFPDD parasites cultured in presence or absence of Shield-1, most PfCK2 -GFPDD-depleted gametocytes failed to develop into mature stage V gametocytes (day 11), even after prolonged periods of observation “ supported with a figure similar to that in Supp 7d.

We performed three biological replicate experiments to score and quantify the gametocyte stages in NF54/AP2-G-mScarlet/CK2 -GFPDD populations cultured ON and OFF Shield-1. These data are now shown in Fig. 3b (please see revised Figure 3 above) and the corresponding Results section has been rephrased accordingly (line 161ff.):

“Closer assessment of gametocytogenesis/gametocyte morphology based on three independent gametocyte maturation assays revealed that from day 8 onwards the morphology of PfCK2 α -GFPDD-depleted gametocytes changed considerably, resulting in stage IV-type cells with elongated and pointy tips that clearly differed from did not progress further to adopt the normal/typical stage V morphology (Fig. observed for +Shield-1 control gametocytes on day 11 (Fig. 3b and Supplementary Fig. 5).”

8. Western blot in Supp 7b. The loading controls are markedly different due to the growth defect in RAPA treatment parasites. Could they provide relative abundance against loading control to determine the quantity of PfCK2a that's here? It's markedly higher than in the in the asexual cKO. Also, when compared to Supp 5a there is more GFP on the blot for cKO than for KD yet the phenotype is so much stronger in the KO. Where is all the protein coming from in the cKO gams? If the protein is lingering, then why isn't it functional? The small amount present in the KD parasites seems adequate to pull them through to Stage IV before they become seriously defective. The reason given for the difference between Tham and this work is more efficient degradation of protein in the 3HADD parasite, but then within this work there are 2 experiments investigating gametocytes and it looks like the stronger reduction in protein level leads to a weaker phenotype. Alongside the different findings between Tham et al and this work, I think this warrants further investigation and discussion.

We agree that this WB leads to believe that residual expression of PfCK2 -GFP in rapamycin-treated NF54::DiCre/CK2 _cKO day 11 gametocytes (Fig. S7) is higher compared to residual CK2 -GFPDD expression in NF54/CK2 -GFPDD KD day 11 gametocytes cultured in absence of Shield-1 (Fig. S5), which is contrary to the expected based on the stronger phenotype observed for the NF54::DiCre/CK2 _cKO gametocytes.

We are grateful to reviewer 1 for pointing this out. This comment made us realise that the WB in Fig. S7b is not informative at all but confusing and misleading instead, and we apologise for not having realised this earlier. The point is that the comparison between the control and rapamycin-treated NF54::DiCre/CK2 _cKO day 11 gametocyte samples essentially compares live stage V gametocytes (control) with dead cells (rapamycin-treated), and hence compares apples with oranges. Based on morphological classification (original Fig. S7d, now Fig. 6b) approx. 3-5% of all parasites in the rapamycin-treated population represent healthy-looking late stage gametocytes (likely parasites with a non-excised *pfck2a* gene and hence normal PfCK2 -GFP and GAPDH expression), while all other cells represent dead/pyknotic/deformed cells that will have largely reduced expression levels of the GAPDH loading control (or of any other protein for that matter). This in turn artificially increases the relative abundance of PfCK2 -GFP compared to GAPDH. Furthermore, due to this profound phenotype the total parasitaemia of NF54::DiCre/CK2 _cKO day 11 cells is approx. 5-fold lower compared to the control (as quantified and reported in the Results section), which required us to harvest much larger culture volumes for the RAPA-treated populations to obtain a decent amount of protein lysate in the first place. For these reasons, this WB result (1) does not and cannot accurately reflect the relative level of PfCK2 -GFP depletion in rapamycin-treated day 11 cells; and (2) is not directly comparable to the other WBs prepared from live parasites of the same line (Fig.

S6c; asexual parasites) or the NF54/CK2⁻-GFPDD KD line [(Figs. S2b (asexual parasites) and S5b (gametocytes)].

We strongly feel that we provided sufficient other convincing data demonstrating that PfCK2⁻-GFP is successfully knocked out in our RAPA-treated NF54::DiCre/CK2⁻ gametocytes. First, we have shown that the rapamycin-induced excision of the *pfck2α* gene in ring stage parasites is highly efficient (Fig. S6). Second, our fluorescence microscopy results clearly demonstrate that the vast majority of rapamycin-treated cells do not express PfCK2⁻-GFP (Fig. 5c; now Fig. 6d). Third, the phenotypic analyses of these gametocytes speak for themselves (Figs. 5 and S7; now Fig. 6). Last but not least, gene KOs are expected to display equal or more pronounced phenotypes compared to protein knockdowns, and this is exactly what our results show – in gametocytes as well as in asexual blood stage parasites.

We therefore removed this misleading WB from the manuscript and adapted the text accordingly (line 268ff.). It is important for us to convey that by no means do we want to hide data by removing this WB. We are simply convinced it is an invalid experiment for the reasons outlined above and we would like to avoid causing the same confusion that reviewer 1 experienced among the readership. If reviewer 1 and the editor disagree with this action, we would be willing to add the WB back to the manuscript and explain that it has to be interpreted with caution and is not very informative.

9. Again, I feel that most of figure Supp7 should appear in the main manuscript. At least c and d but really the whole lot. I personally find this supp figure more interesting than the main Fig 5.

We revised Figs. 5 and S7. The gametocyte PfCK2⁻-GFP localization data (Fig. 5a) remain as a single panel in Fig. 5. The WB data (Fig. S7d) have been removed from the manuscript. Fig. 5b and 5c and all other panels from Fig. S7 are now combined into a new Fig. 6.

Revised Figure 5:

Revised Figure 6:

10. It would be nice to also include a brief description/background of DMAT/TBB as you have done for the other inhibitors. Eg "TTP 22, a cell-permeable ATP analog shown to inhibit the human CK2 kinase *in vitro*".

We now provide brief descriptions for all four inhibitors (line 306ff.).

11. "In summary, we show that three inhibitors of human CK2 of which two showed promising activity against recombinant PfCK2 in *in vitro* kinase assays performed poorly in blocking parasite multiplication in a cell-based assay." I had to reread this to understand it. Perhaps like this?: In summary, we show that three inhibitors of human CK2 performed poorly in blocking parasite multiplication in a cell-based assay, despite two of these having previously shown promising activity against recombinant PfCK2 in *in vitro* kinase assays.

We thank reviewer 1 for this suggestion and changed this sentence accordingly (line 320ff.):

"In summary, we show that ~~three~~four selective inhibitors of human CK2 α performed poorly in blocking parasite multiplication in a cell-based assay, despite two of which~~two of them previously~~ showed promising activity against recombinant PfCK2 α in *in vitro* kinase assays ~~performed poorly in blocking parasite multiplication in a cell-based assay.~~"

a. Also, this is inaccurate. You state TBB was previously tested on PfCK2 α , you've used DMAT, therefore only one of these compounds has truly been tested on PfCK2 α . What was the reason for not using TBB (especially given there are existing IC₅₀ data)? DMAT has improved activity against human CK2 α , this may not carry over to PfCK2 α . Given that you have some activity in DMAT and a structural analog TBB, I think this would be interesting to also test.

We thank reviewer 1 for identifying this mistake that we now corrected in the revised manuscript (line 313ff.):

"Finally, we tested the polyhalogenated benzimidazole compound TBB that blocks recombinant human CK2 α activity with an IC₅₀ of 0.9 μ M⁶⁰ and DMAT, a structural analog of TBB that shows improved activity and selectivity towards human CK2 α ~~compared to TBB⁵⁹⁶¹~~."

We now also tested TBB in our whole cell drug assay and show that it lacks activity against blood stage parasites as well (line 317ff.):

"In our cell-based assay, TBB showed poor activity (IC₅₀ > 50 μ M) and for DMAT we determined an IC₅₀ of 15.8 μ M (17.3 μ M; 14.4 μ M) ~~for DMAT~~ in inhibiting parasite proliferation (Supplementary Fig. S9)."

Revised Figure S9:

Quinalizarin IC50 N/A

TTP 22 IC50 N/A

TBB IC50 N/A

DMAT IC50 15.8 µM

Chloroquine IC50 9.1 nM

Artesunate IC50 6.8 nM

Reviewer #2 (Remarks to the Author)

This manuscript describes the role of protein kinase PfCK2a in the development of human malaria parasite *P. falciparum*. Since this kinase is essential for the blood stage development of the parasite, two different approaches were taken to generate a condition knockout/knockdown to investigate its function. Using mutant parasites, role of PfCK2a was established in asexual as well as sexual development of the parasite. The role of this kinase in asexual development and possibly in invasion has previously been reported (Tham et al. Plos Pathogens 2015). Therefore, the major novel finding in this study is its involvement in gametocyte development. The following issues need to be addressed:

1. The FKBP-DD based approach to knockdown PfCK2a did not alter the asexual development. Using a similar approach, Tham et al., found major defect in parasite growth/invasion. How can these differences be explained? Could it be the different background strain used in the two studies? This and other possibilities should be discussed. Given that there is no phenotype in asexual stages and only a modest difference in late stage gametocyte difference, the data related to FKBP-GFP-DD should be removed or moved to the supplement.

Regarding the difference in performance of the FKBP/DD-dependent knockdown of PfCK2 -HADD vs PfCK2 -GFPDD expression please see our response to comment 3 from reviewer 1. We would prefer keeping the data obtained with this line in the main manuscript as we think they are important in the context of the study. However, if still requested we would propose to move Fig. 2 (data on asexual parasites) to the Supplementary Information, while keeping the gametocyte data in the main manuscript (which we think demonstrate a striking rather than a modest phenotype).

2. The KO using diCre/loxP resulted in major defects in parasitemia in second and subsequent cycle and authors have attributed this to defects in invasion. However, they have not performed direct assays to assess invasion and egress. It is also important to quantify various intraerythrocytic developmental stages to rule out the defects on IDC. In addition, the number of merozoites formed should also be determined, which is important to assess the role in parasite replication especially when this kinase is present in the nucleus.

As reviewer 2 mentioned above, Tham et al. (PLoS Pathogens 2015) already investigated the role of PfCK2 in asexual parasite development and invasion. In their study, parasite progression through the IDC was assessed in control and PfCK2 -HADD-depleted populations by flow cytometry analysis of SYBR Green-stained parasites. These experiments suggested that PfCK2 -HADD-depleted parasites have no apparent developmental defect during intra-erythrocytic development and merozoite egress and that PfCK2 -HADD-depleted merozoites fail to invade RBCs. Our results obtained with the DiCre-inducible PfCK2 KO parasite line confirms this PfCK2 loss-of-function phenotype (Fig. 4). We agree with reviewer 2, however, that neither the Tham et al. study nor our work assessed a potential role for PfCK2 during intra-erythrocytic development and included direct assays to study egress and invasion. We therefore performed additional flow cytometry and microscopy experiments and now demonstrate that PfCK2 KO parasites show delayed progression through schizogony but are still able to complete schizogony and release merozoites. These new results are now presented in the corresponding Results section (line 233ff.) and in Figs. S7 (Results) and S8 (flow cytometry gating strategy).

New Figure S7:

New Figure S8:

3. Did CK2 inhibitors block gametocytogenesis? The efficacy of these inhibitors should also be tested after PfCK2a depletion.

We now also tested TBB against asexual blood stage parasites (please see our response to comment 11 from reviewer 1, and Figure S9 above).

As all four compounds were inactive against asexual stages we don't understand the rationale for why they should be tested against gametocytes or against parasites after PfCK2 -GFPDD depletion. Furthermore, testing compounds for activity against the non-proliferating gametocyte stages requires entirely different drug assays that are not available in our laboratory. In absence of any well-grounded evidence that these human CK2 inhibitors may specifically be active against gametocytes, we feel that establishing and running such an assay for these inhibitors is out of scope of this work.

REVIEWERS' COMMENTS:

Reviewer #1 (Remarks to the Author):

I am satisfied that the authors have very adequately addressed all comments on the manuscript and I congratulate Hitz, Voss and colleagues on a solid piece of work.

I still find it unusual that there is such a large difference in growth between the Tham KD and theirs, however, all their other data support and build on the same phenotype as previously published. This discrepancy between slightly different parasite lines is discussed, and I agree with their response that it would be unnecessary overkill to really dig into this any further.

My confusion in comment 8 is well explained and I agree that including this Western does no more than muddy the waters. I see no way this could be adequately controlled. It could be included with the explanation provided as a supplementary figure to account for the handful of mature stage V seen in Fig. 6b, but I have no strong opinion either way.

Very minor comment

Fig 3d. There are no true stage V in the -shield condition. Perhaps day of gametocytogenesis is a more accurate label here. Also, as gams were used on day 7 not 6, this is more likely a mix of III/IV than IIIs alone.

Reviewer #2 (Remarks to the Author):

Authors have addressed most of my queries.

Author response to reviewers' comments on the revised manuscript "The catalytic subunit of Plasmodium falciparum casein kinase 2 is essential for gametocytogenesis" by Hitz et al.

Reviewer #1 (Remarks to the Author):

I am satisfied that the authors have very adequately addressed all comments on the manuscript and I congratulate Hitz, Voss and colleagues on a solid piece of work.

>>>We thank reviewer 1 for this comment.

I still find it unusual that there is such a large difference in growth between the Tham KD and theirs, however, all their other data support and build on the same phenotype as previously published. This discrepancy between slightly different parasite lines is discussed, and I agree with their response that it would be unnecessary overkill to really dig into this any further.

>>>We thank reviewer 1 for this comment.

My confusion in comment 8 is well explained and I agree that including this Western does no more than muddy the waters. I see no way this could be adequately controlled. It could be included with the explanation provided as a supplementary figure to account for the handful of mature stage V seen in Fig. 6b, but I have no strong opinion either way.

>>>We thank reviewer 1 for sharing our opinion and decided not to include this misleading and non-informative Western blot.

Very minor comment

Fig 3d. There are no true stage V in the -shield condition. Perhaps day of gametocytogenesis is a more accurate label here. Also, as gams were used on day 7 not 6, this is more likely a mix of III/IV than IIIs alone.

>>>We agree with reviewer 1 and changed the labeling in Figure 3d accordingly.

Reviewer #2 (Remarks to the Author):

Authors have addressed most of my queries.

>>>We thank reviewer 2 for this comment.